# Changing Abundance and Distribution of the Wintering Swan Goose *Anser cygnoides* in the Middle and Lower Yangtze River Floodplain: An Investigation Combining a Field Survey with Satellite Telemetry

**An An** [1,2], **Lei Cao** [1,2,*], **Qiang Jia** [3], **Xin Wang** [1], **Qin Zhu** [3], **Junjian Zhang** [1,2], **Xueqin Ye** [1,2] **and Dali Gao** [4]

1   State Key Laboratory of Urban and Regional Ecology, Research Center for Eco-Environmental Sciences, Chinese Academy of Sciences, Beijing 100088, China; anan026@sina.cn (A.A.); wangxin.ustc@gmail.com (X.W.); zhangjunjian1993@163.com (J.Z.); yexueqin14@mail.ucas.ac.cn (X.Y.)
2   College of Resources and Environment; University of Chinese Academy of Sciences, Beijing 100049, China
3   School of Life Science, University of Science and Technology of China, Hefei 230026, China; jiaqiang@mail.ustc.edu.cn (Q.J.); qinzhu9@mail.ustc.edu.cn (Q.Z.)
4   Authority of East Dongting Lake National Nature Reserve, Yueyang 414000, China; daligao@163.com
*   Correspondence: leicao@rcees.ac.cn; Tel.: +86-10-6284-9161

**Abstract:** Migratory waterbird communities are quick to respond to ecosystem degradation, and they are widely considered to be important bioindicators of complex environmental changes. The swan goose (*Anser cygnoides*) has been listed as a globally vulnerable species in the International Union for Conservation of Nature (IUCN) Red List of Threatened Species. This species currently winters almost exclusively in China and is mostly concentrated on lakes in the middle and lower catchment of the Yangtze River floodplain, especially in Poyang Lake, Jiangxi province and some sites in Anhui province. In the past few years, the population of *A. cygnoides* has fluctuated. To protect this fragile Anatidae species, long-term and accurate population estimation is both necessary and urgent. In this study, we evaluated the change in numbers and distribution of *A. cygnoides* by comparing surveys conducted in 2004 and 2005 with more recent ones conducted in 2015 and 2016. A reduction in the count number of this species occurred in the survey sites. After a statistical Mann-Whitney U test, the count numbers of *A. cygnoides* decreased significantly at the survey sites in Anhui province and the abundance decrease at the survey sites in Poyang Lake was only marginally significant. The inaccessibility of the new sites revealed by satellite tracking impeded a more prudent and comprehensive estimate of the population change. Satellite tracking technology may be a tool to consider for increasing the efficiency of data acquisition. Information transmitted from satellite tracking devices can help us to better understand the species' behavior and wintering habitat. This technology has the potential to substitute costly and time-consuming field surveys. Conservation designs and management plans must be created for specific national nature reserves and key wintering sites. A more efficient long-term species monitoring system with improved spatial coverage should be conducted to safeguard wintering *A. cygnoides*.

**Keywords:** swan goose; satellite telemetry; distribution changes; population abundance; Poyang Lake; Yangtze River floodplain

## 1. Introduction

Migratory waterbird communities are quick to respond to ecosystem degradation and they are widely considered to be important bioindicators of complex environmental changes [1–3]. Compared to other migratory waterbirds, birdwatchers are typically drawn to large Anatidae species due to their large population numbers during migration. The census on the abundance and distribution of Anatidae species is relatively easy compared to other rare species and is crucial for both ecologists and species conservation managers in identifying the environmental status of ecosystems [4].

The swan goose (*Anser cygnoides*) belongs to the large Anatidae family and has been listed as a globally vulnerable species in the International Union for Conservation of Nature Red List of Threatened Species [5]. Previous studies have shown that *A. cygnoides* breeds in central and eastern Mongolia, the border regions of China and the Russian Far East. Although *A. cygnoides* formerly wintered in Japan and Korea, it has not been regularly recorded in these countries since the 1950s [6]. This species currently winters almost exclusively in China and is mostly concentrated on lakes in the middle and lower catchment of the Yangtze River floodplain, especially in Poyang Lake, Jiangxi Province and some sites in Anhui province [7–10].

In recent years, the population of *A. cygnoides* has fluctuated. The count data from yearly field surveys conducted in breeding areas in both Mongolia and Russia [11–13] and in wintering areas in China show a decrease in abundance [14]. Compared to other Anatidae species, such as the greater white-fronted goose (*Anser albifrons*) and the bean goose (*Anser fabalis*), the *A. cygnoides* population has decreased steeply [15,16]. The reasons behind this population decrease include wetland degradation, food source shortages and hydraulic changes in wintering areas [14,17]. To protect this fragile Anatidae species, long-term and accurate population estimation and monitoring procedures are both necessary and urgent.

Until recently, the count data derived from field surveys were the primary data source for species population estimation and distribution confirmation. Such surveys were typically costly and time-consuming. In some developed countries (e.g., in Canada), bird monitoring databases are built by the collaboration of scientists, volunteers, government agencies and the general public, which follow well-organized, large-scale survey initiatives [18]. In developing countries, such as China, the contribution of citizen science towards species census still needs specialization [19]. Thus, extensive bird surveys deployed by a combined initiative of professional scientific groups and communities remain uncommon. Additionally, the count data quality is heavily reliant on many factors, such as investigation timing to avoid omissions, observation sites, long-term financial support and skill level of observers. Although field protocols and statistical models have been extensively used to maximize high-quality monitoring data collection [20–24], it continues to be difficult in China to standardize the accuracy and quality of survey data due to different geomorphology and climate situations from lake to lake. Therefore, a modern technology-based survey method, such as satellite telemetry, could be introduced to improve the efficiency and lower the cost of bird surveys in the field. This technique has already been widely used in animal behavior studies and movement ecology research [25–27].

In this study, we report our newly-acquired census results from field surveys and compare them with historical data on numbers and distribution changes of wintering *A. cygnoides* in the middle and lower Yangtze River floodplain. Furthermore, we compare our field survey results in Poyang Lake to our satellite tracking data in order to test a new distribution survey method by using satellite telemetry, which may be applied to reduce expenses, particularly the labor costs required by winter field surveys. The basic information from this study may be useful in further advancing the conservation prospects of this species.

## 2. Study Area and Research Methods

### 2.1. Study Area

The middle and lower reaches of the Yangtze River floodplain start at the Three Gorges Dam, crossing through Hubei, Hunan, Jiangxi, Anhui and Jiangsu provinces to the Yangtze estuary at Shanghai and stretching for 1850 km [15]. The large annual water recession that takes place between summer floods and the winter dry season creates numerous seasonal productive shallow wetlands that together form a large area of approximately 25,000 km$^2$ [28].

Poyang Lake (5100 km$^2$) is the largest freshwater lake in China. It is located in Jiangxi province, which covers the area south of the middle and lower Yangtze River catchments (28°22′ N to 29°45′ N, 115°47′ E to 117°45′ E, Figure 1). This lake undergoes seasonal recession, following the water flow as it moves down the Yangtze River. Each year, the water levels vary from 5 to 22 m (measured at Wu Song) [29]. During the winter dry season, natural drainage exposes large mudflats and independent sublake systems [30]. These dramatic hydrological changes directly affect the characteristics of different habitats and biodiversity [31]. Poyang Lake is rich in aquatic and submerged plants that are an important food source for *A. cygnoides*. Every winter, these migratory waterbirds forage in the herbaceous rich wetlands and shallow waters of this area [17,32]. In recent years, along the middle and lower Yangtze River floodplain, the submerged plant communities have been degraded due to massive aquiculture and this has led to the loss of some important wintering sites, such as Shengjin Lake in Anhui province [14].

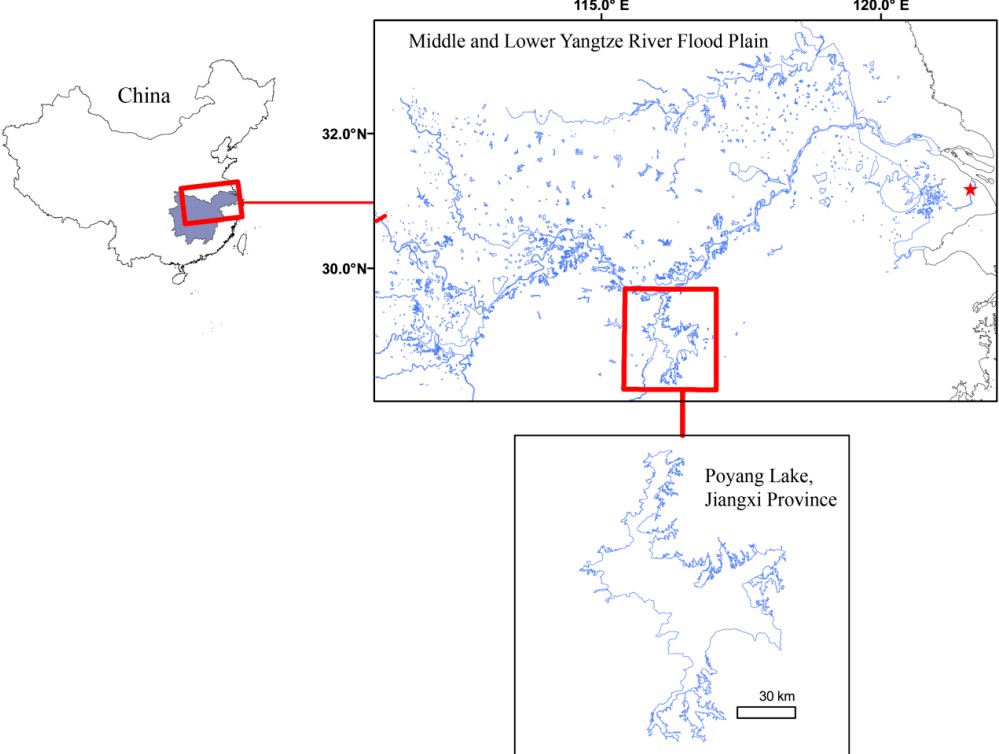

**Figure 1.** Geographic location of the middle and lower reaches of the Yangtze River flood plain and Poyang Lake, China.

### 2.2. Historical Sources and Investigative Data

The first comprehensive and synchronous water bird surveys in this region were conducted in the middle and lower reaches of the Yangtze River floodplain in January and February 2004 and in February 2005 [8,9]. Following this, the authors of this study organized annual field surveys between 2006 and 2012, with mainly the lakes of Anhui province being surveyed (Table A1). A recent investigation

in 2015 and 2016 mainly surveyed lakes in Anhui and Jiangxi provinces. In the investigation in 2016, more provinces (including Hunan, Hubei, Jiangsu and Shanghai) were included to improve the expanse of the investigative area in the middle and lower Yangtze River floodplain. We compared the count data between 2004, 2005, 2015 and 2016 due to the higher quality of the datasets.

For the recent surveys in 2015 and 2016, the observation sites were selected according to the international importance of the Ramsar wetlands [33,34] and former surveys in 2004 and 2005 (Figure 2). The survey period was from December to mid-February the following year. Two or four teams started the surveys at a similar date, with each province having its own team. As the most important wintering area of *A. cygnoides*, Poyang Lake (Jiangxi province) became the primary investigation site. Sixty-three sublakes of Poyang lake were investigated in 2015 as well as an additional 17 sublakes in 2016 (in total 80 sublakes). The recording methodology used was a systematic scheme for the waterbird survey, which was also used by a previous study in Shengjin Lake [21]. The observation sites were designed to divide each sublake into several discrete survey areas using clear geographic boundaries. Individual numbers of a population were counted and recorded using a monocular telescope. Count numbers and the geographic coordinates of observation sites were recorded. Those data were used to compare with historical survey results in 2004 and 2005. To assess whether there were significant differences in bird numbers between the two time periods (2004/5 and 2015/6), the mean values from the two-year data were analyzed using a Mann-Whitney *U* test.

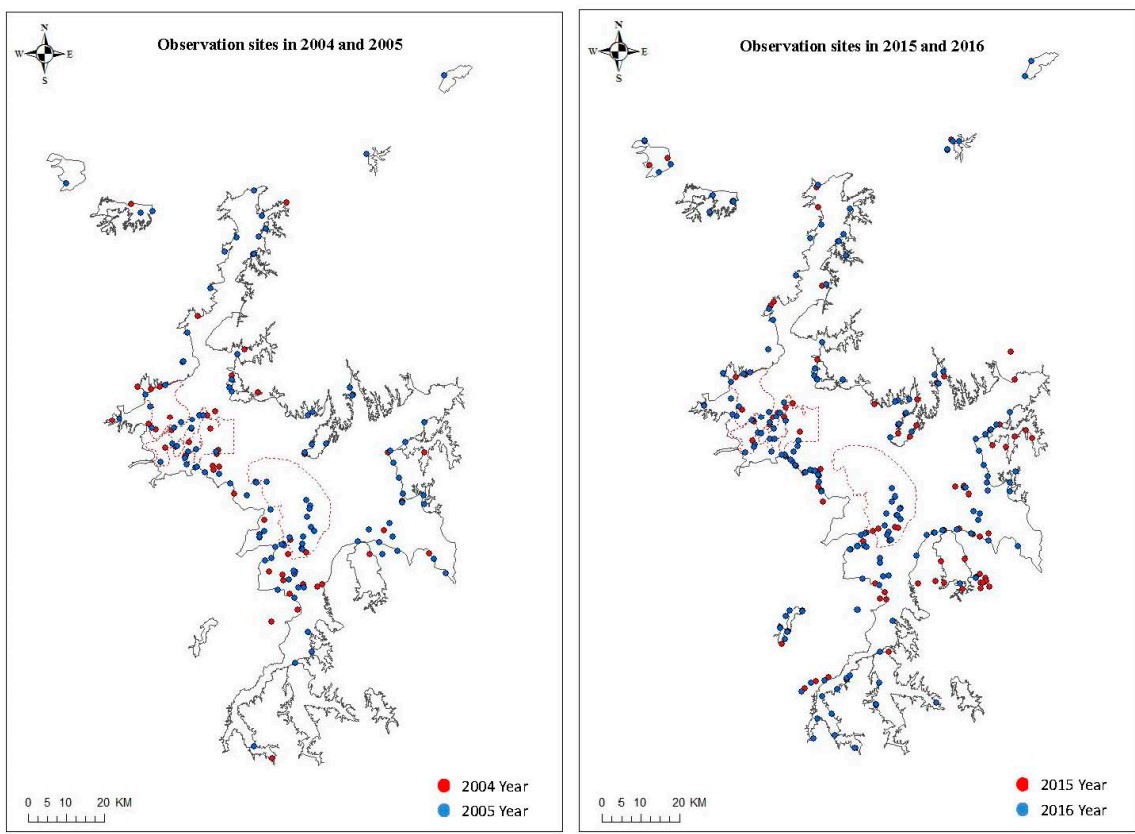

**Figure 2.** Observation sites during the 2004, 2005, 2015 2016 surveys in Poyang Lake.

## 2.3. Satellite Tracking Data

Fifty individual swan geese (*A. cygnoides*) were captured and tagged with satellite tracking loggers at their breeding areas in Mongolia during the molting summer of 2014. After the migration journey, due to a high mortality and signal loss, four of these animals (transmitter codes: SWAG01, SWAG04, SWAG05 and SWAG11) arrived at Poyang Lake and wintered there during the 2014/5 winter season. Three of them were juveniles and one was an adult (Table A2. All of them were fitted with

necklace collar type solar-powered GPS/GSM satellite transmitters from Ecotone Telemetry Company (ecotone-telemetry.com). The total weight of the transmitter equipment was 44 g and we ensured that the weight did not exceed 3% of the total body mass of each bird, which is assumed to minimize any interference in their ability to fly [35]. These satellite telemetry devices were set to record the locations of each bird with GPS accuracy between 00:00 and 22:00 at 2 h intervals every day throughout the research. Data were downloaded via a GPS/GSM platform of Ecotone. The tracking data related to the activities of these birds who wintered in Poyang Lake during the 2014/5 winter season were used in this study. Furthermore, in the following year, SWAG04 (juvenile, male) wintered at Dongting Lake (Hunan province), while the other birds continued to winter at Poyang Lake. The duration of the tracking periods varied from 115 d in 2015 (from 2 December 2014 to 25 March 2015) to 146 d in 2016 (from 31 October 2015 to 25 March 2016). The arrival and departure dates at Poyang Lake were unique to each individual bird and varied from one year to the next for each of them. The mean wintering period of these four tracked *A. cygnoides* varied from 95.67 days to 128.67 days during the 2014/5 and 2015/6 winter, respectively (Table A2). The geographical coordinates generated by those tracking data were used to describe the wintering movements of *A. cygnoides* and we created a satellite tracking map to show the wintering distribution of these birds at Poyang Lake in 2015 and 2016.

## 3. Results

### 3.1. Changing Numbers and Distribution of Anser Cygnoides in the Yangtze River Floodplain

We counted approximately 61,000 *A. cygnoides* in the Yangtze River floodplain in the 2004 and 2005 winter seasons. Anhui and Jiangxi provinces had the highest number of *A. cygnoides*; namely, 95% of total *A. cygnoides* number. In the 2015 and 2016 winter seasons, we counted approximately 8800 and 25,000 *A. cygnoides*, respectively. The vast majority of the birds was concentrated in Jiangxi province, especially in the area of Poyang Lake. The mean counting number of the Yangtze River floodplain showed a 72% decrease in the ten-year period between 2004/2005 and 2015/2016. Furthermore, Anhui province contributed 78% of the total 72% decrease in counting number (Table 1).

**Table 1.** Total count data of *A. cygnoides* in the middle and lower reaches of the Yangtze River floodplain per province during 2004, 2005, 2015 and 2016 (count taken in January or February).

| Years | Anhui | Jiangxi | Hubei | Hunan | Jiangsu | Shanghai | Total |
|---|---|---|---|---|---|---|---|
| 2004 | 31,197 [a] | 29,378 [a] | 139 [a] | 172 [a] | 0 [a] | 0 [a] | 60,886 |
| 2005 | 37,519 [b] | 22,313 [b] | 1260 [b] | 76 [b] | 0 [b] | 10 [b] | 61,178 |
| Average of 2004&2005 | 34,358 | 25,846 | 700 | 124 | 0 | 5 | 61,032 |
| 2015 | 67 [c] | 8751 [c] | N.A. | 80 [d] | N.A. | N.A. | 8898 |
| 2016 | 190 [e] | 24,987 [e] | 22 [e] | 37 [e] | 0 [e] | 0 [e] | 25,236 |
| Average of 2015&2016 | 129 | 16,869 | 22 | 59 | 0 | 0 | 17,067 |

[a] Barter et al. (2004); [b] Barter et al. (2006); [c] January and February count in 2015; [d] Winter survey from the Authority of East Dongting Lake National Nature Reserve, China (2015); [e] January and February count in 2016. N.A. = No data available, N.A. values were not counted in during average calculation.

In Anhui province, 11 and 9 lakes were investigated in 2004 and 2005, respectively. Only 4 of them were identified as key sites for *A. cygnoides* (*A. cygnoides* > 550, Waterbird Population Estimation 3rd Edition, 2002, Wetland International, Wageningen, The Netherlands), where the counting numbers were 30,880 (in 2004) and 37,030 (in 2005). Nearly 98% of *A. cygnoides* wintered at these 4 key sites in Anhui province with mean bird numbers of 7720 (in 2004) and 9258 (in 2005) per site. A total of 8 and 9 lakes were investigated during the 2015 and 2016 winter, respectively. In total, 67 and 190 *A. cygnoides* were counted at those lakes with no sites identified as key sites of importance for *A. cygnoides* (*A. cygnoides* < 680, Waterbird Population Estimation 5th Edition, 2012). Over the last 10 years, all lakes in Anhui province lost their designation of being internationally important wetlands for this species, including Shengjin Lake, Baidang Lake, Fengsha Lake, Caizi Lake and Wuchang Lake (Tables 1 and 2).

In Poyang Lake, 52 and 72 sublakes were investigated in 2004 and 2005, respectively. However, only 4 and 6 of these sublakes were identified as key sites, where the total numbers of this species were 28,687 (in 2004) and 21,205 (in 2005). During the 2015 and 2016 winter, we investigated 63 and 80 sublakes in Poyang Lake, respectively. In total, we recorded 33,738 *A. cygnoides* in Poyang Lake throughout a 2-year period, including 8751 in 2015 and 24,987 in 2016. During the 2015 and 2016 investigation, a large proportion of the counted birds of *A. cygnoides* was concentrated in 6 and 8 key sites and the centralization rate dropped from 97.65% and 95.03% to 84.31% and 85.12%. The mean count number per key site dropped from 7172 (in 2004) and 3534 (in 2005) to 1230 (in 2015) and 2659 (in 2016), respectively.

The count numbers showed a statistically significant decrease over the ten years in Anhui province as tested by a Mann-Whitney *U* test ($p = 0.010$). On the contrary, the number decreases in Poyang Lake were only marginally significant ($p = 0.063$; Table 2).

**Table 2.** Abundance changes of *A. cygnoides* in lakes and key sites of Anhui province and Poyang Lake over ten years.

| Area | Year | Lakes | Total Count | Key Sites | Count at key Sites [a] | Ratio (%) [b] | Mean Count in Key Sites | S.D. in Key Sites | *p* Value [c] |
|---|---|---|---|---|---|---|---|---|---|
| Anhui Lakes | 2004 | 11 | 31,197 | 4 | 30,880 | 98.98 | 7720 | 4614 | |
| | 2005 | 9 | 37,519 | 4 | 37,030 | 98.70 | 9258 | 5145 | 0.010 |
| | 2015 | 8 | 67 | 0 | 0 | 0 | 0 | 0 | (Mann-Whitney *U*) |
| | 2016 | 9 | 190 | 0 | 0 | 0 | 0 | 0 | |
| Poyang Lake | 2004 | 52 | 29,378 | 4 | 28,687 | 97.65 | 7172 | 8689 | |
| | 2005 | 72 | 22,313 | 6 | 21,205 | 95.03 | 3534 | 1827 | 0.063 |
| | 2015 | 63 | 8751 | 6 | 7378 | 84.31 | 1230 | 462 | (Mann-Whitney *U*) |
| | 2016 | 80 | 24,987 | 8 | 21,270 | 85.12 | 2659 | 1212 | |

(a) The 1% international population threshold is a key standard to identify the importance (key site) of a wetland influencing a species. This threshold varied by the fluctuation of the international population estimation and it was at 550 in 2004 and 2005 (WPE 3), and at 680 in 2015 and 2016 (WPE 5). (b) Ratio (%) represents the centralization level at key sites. (c) *p*-value represents the significance level with the Mann-Whitney *U* test: Marginally significant at $p < 0.1$; and significant at $p < 0.05$.

The changes in the distribution of *A. cygnoides* during 2015–2016 compared to those 11 years earlier are shown in the distribution maps of Figure 3. Previously, sublakes Jishanhu and Xinmiaohu (Duchang County, the Jiujiang City area, Jiangxi province) that are situated north of Poyang Lake supported more than 550 birds during the wintering period of 2004 and 2005. However, fewer than 50 birds were counted in these two sites during the 2015 and 2016 wintering season. On the other hand, 2104 birds were counted in February 2015 and 3859 in January 2016 at sublakes Dalianzihu and Nanjianghu, respectively (Poyang County, Shangrao City, Jiangxi province, China), which are located near the Kangshan Dam south of Poyang Lake. These two sublakes subsequently became new international key sites for this species due to the 2015 and 2016 censuses. Wintering *A. cygnoides* inside the Poyang Lake National Nature Reserve also showed slight changes in their distribution. For example, sublakes Banghu, Changhuchi and Dahuchi used to be the primary wintering areas for *A. cygnoides*, but they were replaced by sublakes Zhonghuchi and Dachahu by 2015. Progressively more *A. cygnoides* started to move to the Nanji Wetland National Nature Reserve and winter there. By the winter of 2015, two sublakes within this national nature reserve contained over 680 birds of the *A. cygnoides*. Furthermore, sublake Linchonghu used to be a very important wintering area for *A. cygnoides* in 2005; this lies on the west coast of Poyang Lake. However, none of those sites sustained individual numbers above 680 by 2015 and 2016 (Figure 3, Figure A1 and Table A3).

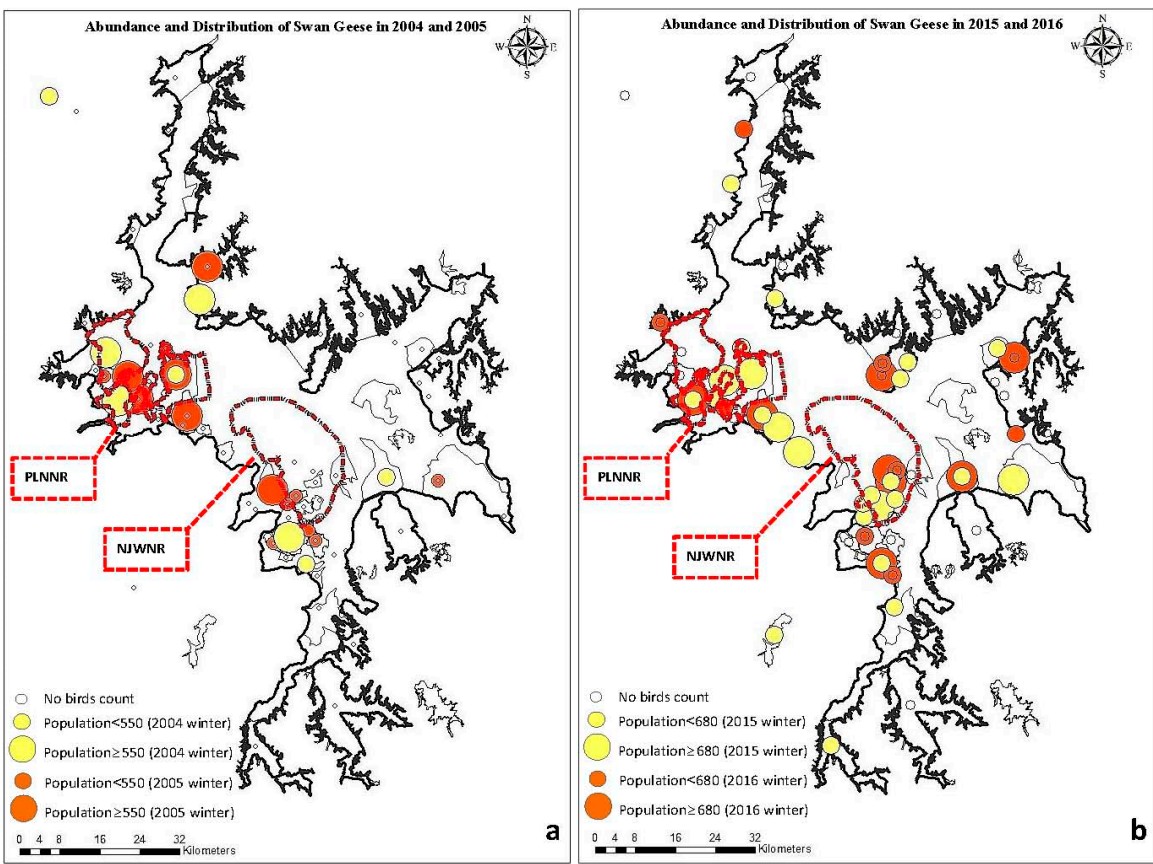

**Figure 3.** Abundance and distribution of *A. cygnoides* in Poyang Lake in the winters of (**a**) 2004 and 2005 and in the winters of (**b**) 2015 and 2016. Abbreviations: PLNNR: Poyang Lake National Nature Reserve; NJWNR: Nanji Wetland National Nature Reserve.

## *3.2. Wintering Tracking Using Satellite Telemetry*

Based on the satellite signals we received, 637 and 3340 geographic coordinates were recorded to reveal wintering activities of these four *A. cygnoides* though the winters of 2015 and 2016, respectively. The major wintering areas were identified by dense satellite tracking signals for these tracked animals, including Dachahu sublake (Poyang Lake National Nature Reserve, SWAG05); Candouhu, Xiaotanhu and Dawuhu sublakes (SWAG01, SWAG04 and SWAG11); Sanniwan and Baishahu sublakes (Nanji Wetland National Nature Reserve, SWAG01, SWAG04 and SWAG11); and the mudflat area, including sublakes Nanjianghu and Dalianzihu (SWAG01 and SWAG04) in the 2014/5 winter (Figure 4a). Nanji Wetland National Nature Reserve became the most important wintering area for these *A. cygnoides* in the 2015/6 winter. Apart from one missing bird (SWAG04), the other birds spent nearly the entire wintering period (mean wintering period at 128.67 days, Table A2) in the sublakes of Sanniwan, Baishahu and Zhanbeihu, which are located in the Nanji Wetland National Nature Reserve (Figure A1). SWAG 01 first arrived at Dawuhu sublake on October 2015 and stayed there for approximately 16.8 d (Figure 4b, Table A2 and Figure A1). SWAG11 stayed for a short period (12.2 d) in Jinxihu and Chengjiahu sublakes later in February 2016 before its spring migration (Figure 4b).

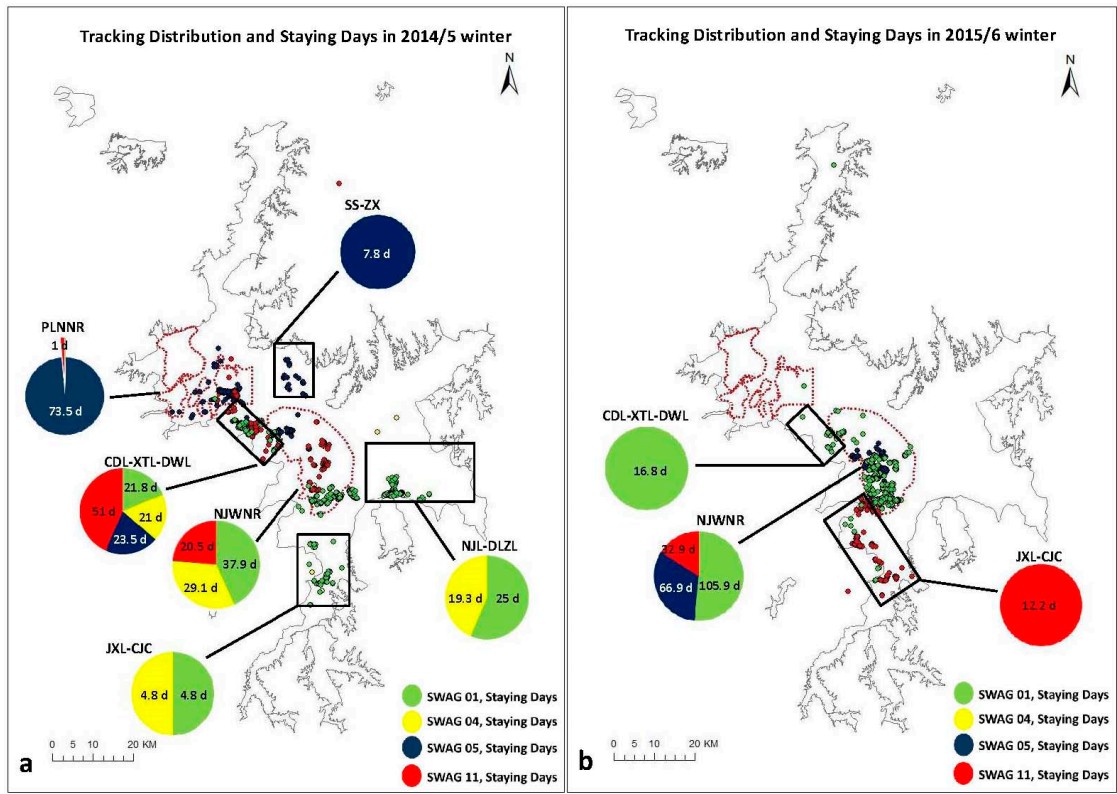

**Figure 4.** Staying days in each distributed area of tracked *A. cygnoides* in Poyang Lake in 2014/5 and 2015/6 winters. (**a**) Activity areas for the tracked *A. cygnoides* in 2014/5 winter. (**b**) Activity areas for the tracked *A. cygnoides* in the 2015/6 winter. Staying days were calculated by using time intervals of continuous satellite tracking signals. Abbreviations: PLNNR: Poyang Lake National Nature Reserve; CDL-XTL-DWL: Mudflat areas in sublakes Candouhu, Xiaotanhu and Dawuhu; NJWNR: Nanji Wetland National Nature Reserve; JXL-CJC: Sublakes, including Jinxihu and Chengjiachi; NJL-DLZL: Mudflat areas in sublakes Nanjianghu and Dalianzihu; SS-ZX: Mudflat areas in Sishan (Huamiaohu) and Zhouxi.

## 4. Discussion

Vulnerable species, such as *A. cygnoides*, can serve as good ecological indicators of the health status of wetland biodiversity. Currently, field surveys conducted by humans are the primary method for studying the abundance and distribution of *A. cygnoides*. Species population and distribution changes may reveal deeper consequences related to environmental changes. As shown in Table 1, the abundance of *A. cygnoides* in the middle and lower reaches of the Yangtze River floodplain dropped in terms of count numbers over the 12-year period (2004–2016). Consistent with other studies [14,36], our results showed that wetland habitats in Anhui and Jiangxi provinces held the majority of the last wintering *A. cygnoides*. It is most worrisome that lakes in Anhui province, which are former Ramsar wetlands, are losing their importance for wintering *A. cygnoides* [33,34]. Although census-based surveys provide a quantity of data, the quality still needs to be improved. For example, in Table 2, the count numbers in Poyang Lake fluctuated dramatically between 2015 and 2016. Weather conditions and experiences of different observers between years may cause such a huge count difference.

An earlier study in 2012 recognized important sites for *A. cygnoides*, including the Poyang Lake National Nature Reserve, the Nanji Wetland National Nature Reserve, Poyang County and the Duchang Nature Reserve [37]. However, according to our recent field surveys, obvious distribution changes occurred in the northeast coast areas of Poyang Lake. One important wintering site in the north of Poyang Lake disappeared (Figure 3a,b). This could have resulted from the aquacultural and tourism activities taking place in the north counties in recent years. Furthermore, we have shown that a greater

proportion of the *A. cygnoides* population was distributed to a new wintering area in the middle of the Poyang Lake National Nature Reserve and the Nanji Wetland National Nature Reserve in the 2015/2016 winters. This means that the Candouhu–Xiaotanhu–Dawuhu sublake region is becoming a new wintering destination of this species. On the contrary, *A. cygnoides* were less attracted to the Poyang Lake National Nature Reserve, which was possibly due to the rapid development in tourism and road construction in the area. Although there are many studies on environmental responses, studies focusing on the underlying mechanisms behind the change in the winter habitat of *A. cygnoides* and the impact of anthropogenic disturbance on abundance of *A. cygnoides* are recommended.

Food availability may be the primary reason behind the distributional change of this tuber feeding waterbird [14,38]. Previous studies have shown that the habitat area, littoral zone slopes, weather conditions, vegetation availability and protection status positively affect grazing waterbirds [38–40]. Local aquaculture and hydrological processes may influence the annual yields of submerged plants while also affecting the distribution of tuber grazing geese species [14,41]. *Vallisneria spiralis* is a major food source for *A. cygnoides*, which can be accessed from muddy lake sediments after water recedes [42]. This submerged plant is a dominant species in the middle and lower reaches of the Yangtze River floodplain. Studies on this species have shown that fluctuations in water levels significantly affect the growth and distribution of this plant [43]. In 2014, a decrease in the density and biomass of this submerged macrophyte was reported in Dahuchi Lake, Meixihu Lake, Shahu Lake and Sixiahu Lake, which are located north of the Poyang Lake National Nature Reserve [44]. High water levels may have induced the decrease in these submerged plants. The high water level in Dahuchi Lake was also maintained at 15 m from January 2014 to March 2015 [45]. Furthermore, a reduction in food availability may have caused a decrease in the frequency of this species visiting wintering areas, such as Dahuchi Lake, south of the Poyang Lake National Nature Reserve and Dachahu Lake, and even promoted the southern migration to a new settlement area in the Xiaotanhu and Dawuhu sublake region. Thus, high water levels might have influenced food availability for *A. cygnoides*. In this study, an example was found in new formed wintering areas. Dalianzihu, Nanjianghu and Kangshanhu belong to the same sublake region. The Kangshan Dam separates the area as Dalianzihu and Nanjianghu to the north (Figure A1, No. 24) and Kangshanhu to the south (Figure A1, No.26). The water level at Kangshanhu remains high and it is subsequently used as a reservoir. Thus, compared to Kangshanhu, the mudflats in sublakes Dalianzihu and Nanjianghu attracted a large population of *A. cygnoides*. The richness of submerged plants is influenced by many complex environmental factors. Currently, studies on plant flora are only carried out at certain sublakes or reservation areas in Poyang Lake, such as Dahuchi Lake and the Nanji Wetland National Nature Reserve [44,46]. There is still a lack of studies on food web and biological energy flow, especially any focusing on the relationship between plant richness and waterbird distribution.

Satellite tracking, whereby locations of animals are regularly uploaded from satellite transmitters, may be a tool to consider for increasing the efficiency of data acquisition. This technology has the potential to substitute costly and time-consuming field surveys. As shown in the results of Figures 3b and 4a, the satellite tracking data based distribution maps were compatible with the survey data based distribution maps. The key sites presented by survey data in Figure 3b also showed a high density of tracking records in Figure 4a. However, the inaccessibility and low visibility because of tall vegetation and regular mist at these sites, such as Dachahu, Sanniwan, Dawuhu and the central area of Poyang Lake, reduced the actually coverage of the surveys, and therefore we cannot derive a more prudent conclusion on the overall population trend.

Satellite tracking technology has played a very important role in studies of bird migration [25,26,35]. For migratory waterfowl, the knowledge about the ecology of animal movements can be further enhanced though this tool. For instance, information collected on arrival, departure and stopover days can help us to better understand the species' behavior and wintering habitat. We evaluated the change in numbers and distribution of *A. cygnoides* by comparing surveys conducted in 2004 and 2005 with more recent ones of 2015 and 2016.

## 5. Conclusions

The population size of *A. cygnoides* has greatly fluctuated over the years. The decline in the count number of this species occurred at the survey sites. The count numbers of *A. cygnoides* decreased significantly at the survey sites in Anhui province, whereas the decrease in the abundance of this species at the survey sites in Poyang Lake was only marginally significant.

For the Poyang Lake area, the distribution of *A. cygnoides* changed over the study period. Although the Anhui and Jiangxi provinces had the highest numbers of *A. cygnoides* in 2004 and 2005, almost all counted birds were concentrated only in Jiangxi province in the winter season of 2015 and 2016. New settlement areas for wintering *A. cygnoides* arose in between the Poyang Lake National Nature Reserve and the Nanji Wetland National Nature Reserve. A centralizing trend towards the center of the lake was found by our collar-marked study birds using satellite tracking devices.

Currently, the primary method for studying the abundance and distribution of species uses field censuses, requiring high numbers of man hours. However, inaccessible areas, such as the marsh and muddy area in the center of a lake, limited the observation coverage during our field survey. New technologies, such as unmanned aerial vehicles or satellite tracking, may help to reduce this natural limitation.

Conservation designs and management plans must be created for specific national nature reserves and key wintering sites. A more efficient long-term species monitoring system should be conducted to safeguard wintering *A. cygnoides*.

**Author Contributions:** A.A. and L.C. conceived and designed the study. A.A. organized the field surveys. Q.J., X.W., Q.Z., J.Z., X.Y. and D.G. participated in the field surveys, A.A. analyzed the data and wrote the paper. All authors read and approved the final manuscript.

**Funding:** The study was supported by the National Key Research and Development Program of China (Grant No. 2016YFC0500406), the Chinese Academy of Sciences Key Strategic Program, Water Ecological Security Assessment, the Major Research Strategy for Middle and Lower Yangtze River (Grant No. ZDRW-ZS-2017-3), the National Natural Science Foundation of China (Grant No. 31661143027 and No. 31670424) and China Biodiversity Observation Networks (Sino BON).

**Acknowledgments:** We would sincerely like to thank Fox, A.D., Ely, C., Takekawa, J., Kim, I., Smith, L., Batbayar, N. and all the volunteers for their contributions to the capture of our satellite tracking specimens during the 2014 summer. We would also like to thank the staff members of the Shengjin Lake National Nature Reserve, Poyang Lake National Nature Reserve and the Nanji Wetland National Nature Reserve for their cooperation during our field surveys.

**Conflicts of Interest:** The authors declare no conflict of interest.

## Appendix A

**Table A1.** Swan Geese count data from provinces in the middle and lower Yangtze River floodplain (2004–2016 winters).

| Years | Anhui | Jiangxi | Hubei | Hunan | Jiangsu | Shanghai | Total |
|---|---|---|---|---|---|---|---|
| 2004 | 31,197 [a] | 29,378 [a] | 139 [a] | 172 [a] | 0 [a] | 0 [a] | 60,886 |
| 2005 | 37,519 [b] | 22,313 [b] | 1260 [b] | 76 [b] | 0 [b] | 10 [b] | 61,178 |
| 2006 | N.A. | N.A. | N.A. | N.A. | 174 [c] | 14 [c] | 188 |
| 2008 | 7,861 [c] | N.A. | N.A. | N.A. | N.A. | N.A. | 7861 |
| 2009 | 854 [c] | N.A. | N.A. | N.A. | N.A. | N.A. | 854 |
| 2010 | 1427 [d] | N.A. | N.A. | N.A. | N.A. | N.A. | 1427 |
| 2012 | 3741 [e] | N.A. | N.A. | N.A. | N.A. | N.A. | 3741 |
| 2015 | 67 [f] | 8751 [f] | N.A. | 80 [f] | N.A. | N.A. | 8898 |
| 2016 | 190 [g] | 24,987 [g] | 22 [g] | 37 [g] | 0 [g] | 0 [g] | 25,236 |

[a] Barter et al. (2004); [b] Barter et al. (2006); [c] January count in 2006, 2008, and 2009; [d] February count in 2010; [e] December and January count in 2011/2012; [f] January count in 2015; [g] January and February count in 2016. (N.A. means Not Available).

**Table A2.** Wintering period of satellite tracked Swan Geese throughout 2015–2016.

| Transmitter No. [a] | Age | Sex | Weight (g) | Proportion (%) [b] | Capture sites | Arrival Area | Arrival Time (BJ Time) | Departure Area | Departure Time (BJ Time) | Wintering Period (Days) |
|---|---|---|---|---|---|---|---|---|---|---|
| SWAG01 | Juv | F | 2220 | 1.98 | Gulaat Lake (49.73755 N; 115.30893 E) | Dachahu Lake (PLNNR) | 2014/12/2, 20:00 | Baishahu Lake (NJWNR) | 2015/3/12, 2:00 | 99 |
|  |  |  |  |  |  | Dawuhu Lake | 2015/10/31, 14:01 | Baishahu Lake (NJWNR) | 2016/3/25, 2:00 | 146 |
| SWAG04 | Juv | M | 2050 | 2.14 | Gulaat Lake (49.73755 N; 115.30893 E) | Sanniwan (NJWNR) | 2014/12/4, 18:09 | Nanjianghu Lake | Defect | — |
|  |  |  |  |  |  | Dongting Lake (Hunan) | 2015/12/7, 17:50 | Donting Lake (Hunan) | 2016/3/21, 14:01 | 105 |
| SWAG05 | Juv | M | 2230 | 1.97 | Gulaat Lake (49.73755 N; 115.30893 E) | Zhonghuchi (PLNNR) | 2014/12/5, 2:00 | Sishan and Zhouxi Region | 2015/3/25, 20:00 | 111 |
|  |  |  |  |  |  | Nihu Lake (NJWNR) | 2015/11/26, 2:00 | Nihu Lake (NJWNR) | Defect | — |
| SWAG11 | Ad | M | 3730 | 1.18 | Bus Lake (49.73799 N; 115.1498 E) | Dachahu Lake (PLNNR) | 2014/12/24, 8:00 | Nihu Lake (NJWNR) | 2015/3/12, 8:00 | 77 |
|  |  |  |  |  |  | Baishahu Lake (NJWNR) | 2015/11/26, 17:59 | Baishahu Lake (NJWNR) | 2016/3/11, 8:01 | 135 |

Note: [a] Transmitter mounting type: Neck collar, manufactured by Ecotone Telemetry, Poland (ET_NC); device mass at 44g. [b] Proportion: Telemetry logger weight/Goose Body weight *100%. Abbreviations: PLNNR: Poyang Lake National Nature Reserve; NJWNR: Nanji Wetland National Nature Reserve.

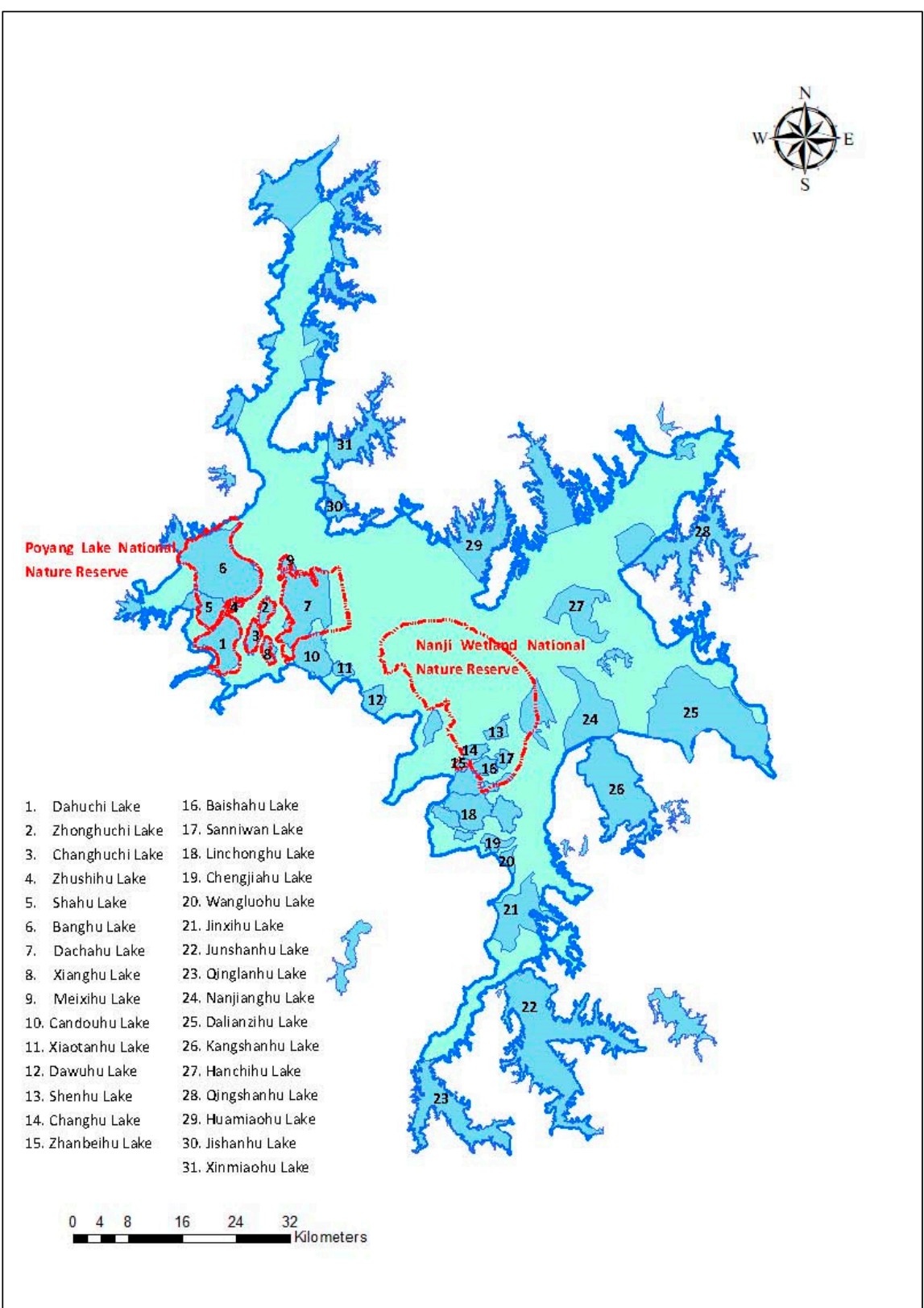

**Figure A1.** Names of major sublakes in Poyang Lake.

**Table A3.** Count numbers of *A.cygnoides* in key sites of Poyang Lake over ten years.

| Key sites | 2004 | 2005 | 2015 | 2016 | Latitude | Longitude |
|---|---|---|---|---|---|---|
| Baishahu Lake | N.A. | 0 | 1137 | 458 | 28.9074 | 116.3043 |
| Banghu Lake | 20,000 | 17 | 0 | 0 | 29.2231 | 115.9315 |
| Beijiahu Lake | N.A. | 0 | N.A. | 1260 | 28.9673 | 116.33064 |
| Candouhu Lake | 0 | 6000 | 88 | 3326 | 29.0941 | 116.0843 |
| Changhu Lake | 0 | 3400 | 190 | 0 | 29.1341 | 115.9873 |
| Chengjiachi Lake | 1 | 18 | 2 | 3020 | 28.8040 | 116.2960 |
| Dachahu Lake | 3 | 3625 | 1164 | 30 | 29.1700 | 116.0712 |
| Dahuchi Lake | 4400 | 222 | 1 | 2130 | 29.1343 | 115.9443 |
| Dalianzihu Lake | 0 | 32 | 2104 | 15 | 28.9261 | 116.5863 |
| Dashafanghu Lake | N.A. | 5000 | N.A. | 0 | 28.9425 | 116.2431 |
| Dawuhu Lake | 0 | 0 | 779 | 45 | 29.0191 | 116.1504 |
| Huamiaohu Lake (Sishan area) | N.A. | N.A. | 0 | 1750 | 29.1408 | 116.3407 |
| Jishanhu Lake | 787 | 0 | 46 | 0 | 29.2991 | 116.1388 |
| Linchonghu Lake | 3500 | 70 | N.A. | 0 | 28.8546 | 116.2677 |
| Nanjianghu Lake | 300 | N.A. | 65 | 3859 | 28.9428 | 116.4824 |
| Shenhu Lake | N.A. | 0 | 4 | 4546 | 28.9482 | 116.3340 |
| Xiaotanhu Lake | 0 | 0 | 932 | 0 | 29.0675 | 116.1150 |
| Xinmiaohu Lake | 0 | 2250 | 0 | 0 | 29.3556 | 116.1617 |
| Zhonghuchi Lake | 0 | 0 | 1262 | 42 | 29.1633 | 116.0144 |
| Zhuhu Lake | 0 | 0 | 0 | 1379 | 29.1443 | 116.6188 |
| Zhushihu Lake | 0 | 930 | 0 | 0 | 29.1764 | 115.9714 |

N.A. means Not Available.

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
