# Peer review of "Changing Abundance and Distribution of the Wintering Swan Goose Anser cygnoides in the Middle and Lower Yangtze River Floodplain: An Investigation Combining a Field Survey with Satellite Telemetry"

_sustainability, doi:10.3390/su11051398_

Round 1
Reviewer 1 Report
This original research paper analyzes the changes in abundance and distribution patterns of the wintering swan goose in China across 10 years in several lakes located the Yangtze River floodplain (China) and relates these changes to environmental degradation in the region.
The merit of this study is to analyze the changes in abundance making use of complementary techniques that include file surveys and radio telemetry. The authors evaluate the observed changes making use likely of all the available literature on previous surveys on the species distribution and on the knowledge of the drivers of change. The changes are also discussed in the light of the international requirement from the Ramsar convention, which supports the recognition of the studied wetlands for the long-term survival of the species.
Given the large number of sub-lakes under study and the relatively limited availability of resources to conduct the survey, I applaud the efforts of the authors to summarize the results. However, I also believe that a higher level of synthesis is certainly needed if the aim of the paper is to compare the distribution range and abundance. The paper will benefit by the introduction of some summarizing statistical indicators (medians and interquartiles across lakes) and by the use of some simple non-parametric tests for the comparison of median and dispersion between the two time steps (like Mann-Whitney U test for medians). This would improve the robustness of the observations that the authors make about the decline of the species.
To summarize, the paper has multiple points of interest for the broad audience of Sustainability but at present, it cannot be accepted in the present form. I recommend its reconsideration after major revision incorporating the points suggested in the attached pdf, which include, among the others, the introduction of statistical indicators and tests as previously discussed.

Author Response
Thanks very much for the review! You are a patient, work carefully and responsibly person. You give me lots of useful suggestions. We thank you for your affirmation and support, and we can't finish our work without your help! It is very nice have such communication with you. We summarized major questions as below, thanks so much again!
Q1. a higher level of synthesis is certainly needed if the aim of the paper is to compare the distribution range and abundance
A: Thank you for your suggestion, we have added more statistical analysis to the significance in decline between years. As your advices, we used Mann-Whitney U test to analysis the significance level of declines in Anhui and Poyang Lake between 10 years gap. The result is, decline in abundance significant in Anhui, and marginally significant in Poyang Lake. We put it in Table 2 as part of our results. Please see Page 6, Table 2.
Q2. Abstract: P1, line 26, This is not very relevant to the topic of the paper.
A: Thank you for your suggestion, we have totally rewritten our abstract based on our revised paper, please see Page1, Line 16-31.
Q3. Introduction: P2,line 2, You may want to cite a more general study here?
A: Thank you for your suggestion, we have changed to some more relevant citations. Please see Page 2, Line 2-3.
Q4. Introduction: P2,line 38, Try to generalize the aim of your study beyond your specific case. Then go into detail and describe how you fulfill this aim through your study case.
A: Thank you for your suggestion, we have rewritten this part in our introduction. Please see Page 2, Line 39-44.
Q5. Methods: P4, Figure 2, add "in the Poyang Lake".
A: Thank you for your suggestion, we have added it in the heading, please see Page 4, Figure 2.
Q6. Methods: P5, Line 5, How many days was the telemetry lasting, from which day to which day?
A: Thank you for your suggestion. The duration of the tracking data used in this study was 115 days in 2015 and 146 days in 2016. We have added this information in our method part. Please see Page 4, Line27-28, Page 5, Line 1.
Q7. Results: P5, Line 16, report the proportion (%) of decline. Provide mean counts and standard deviation (among sublakes) for each year.
A: Thank you for your suggestion. We calculated the mean value of 2004, 2005 and mean value of 2015, 2016. And we calculated the proportion of decline during the two time steps. We have detailed counts in each sub-lakes in Anhui and Poyang Lake, thus, we calculated the mean value of total counts in each sub-lakes in 2004 and 2005, and the same to the total counts in 2015 and 2016 for Anhui and Poyang Lake in Table 2, and then we tested the significant level of decline in Anhui and Poyang Lake respectively by using Mann-Whitney U test. Please see Page 5-6, Table 1 and 2.
Q8. Results: P6, Figure 3, It is not very clear the distinction among the three types of dots. Different size must be report as legend in the map.
A: Thank you for your suggestion, we have changed the colors of the dots, and the meaning of different size has listed as legend in the map, please see Page 6, Figure 3.
Q9. Results: P9, Line 6, what do you mean by distribution activities?
A: Thank you for your asking, the “distribution activities” is not a proper word in English, we would like to explain the changes in movements of those tracking specimens and where they stayed. We have changed this improper word. Please see Page 7,Line 34-48.
Q10. Discussion: P11, Line 21, this is general. Specify.
A: Thank you for your suggestion, we have rewritten our discussion part by a description hierarchical structure as discuss on abundance change, discuss on distribution changes and the possible influencing environmental factors, and the promising future of satellite technology used in replace human-based field work. Please see Page 8- 10.

Reviewer 2 Report
This paper is unclear in its intent and objectives. It explores medium term changes (c.12 years) in swan goose abundance and distribution. There is no statistical analysis. An aim is to compare satellite tracking/remote sensing data as a potential alternative technique addressing concerns about human error in observational work and cost efficiency. However, data for only four birds are presented which seems an insufficient sample size to draw any conclusions. The behaviour of four birds taken at random is highly likely to mirror the behaviour of the population at large due to chance alone. The paper also talks extensively about possible reasons for abundance/distribution changes but there are no/little data/evidence to back these up. I would recommend the authors simplify and streamline this paper focussing just on the population changes, backup up by statistical analysis and concluding with a short discussion of possible causes and recommendations for further work.
Abstract
Line 21: ‘Status of an ecosystem’. Is this the primary objective, what about conserving the bird itself?
Line 24: How does it contribute to environmental health? The authors talk about the bird contributing to ecosystem health but then say changes in birds are caused by ecosystem health. There are many examples of sweeping statements throughout the paper but with a lack of detail to show how they apply to this particular study/example.
Line 25: Clarifying the degree of ecosystem…… This sentence is confused. What is species investigation? How does it clarify the degree of ecosystem degradation, is there even degradation? What biodiversity offsetting are you referring to? This is the only mention of offsetting.
Line 28: ‘…with satellite telemetry.’ To do what? For what purpose?
Line 32: This is very speculative without data to back it up. It should be part of the discussion but not the abstract.
Line 35: The aims of the study should not be at the end of the abstract, it should be the results/implications.
Introduction
Line 4: This sentence does not make sense. How can they be bioindicators of monitoring initiatives? Perhaps used in monitoring initiatives?
Line 5: The relevance of this statement is unclear.
Line 7: And conserving the birds…..?
Line 11: ‘basically distributed soley’ poor grammar.
Line 22: But you listed climate change in the abstract? Are the reasons really known or this all speculation? I think you need to be clearer and consistent.
Line 27: What database?
Line 30: Is a culture of bird watching really something that is ‘developed’?
Line 43: This paragraph appears to be suggesting that most bird data is collected by professionals in China and that the quality is dependent on how much they get paid. Whereas what I think you mean to say is that surveys are generally only carried out by professionals, of which there is a lack, and the time-consuming nature of them means new cost-effective methods are needed.
Methods
P3, line 14: ‘benefit A. cygnoides as a food source’.
P3, line 17: How have they degraded? Fewer species, invasive species, reduced abundance, biomass?
P4, line 2: Of all sites? Where is the 2006-2012 data?
P4, line 3: This sentence is not needed. And what historical data? I presume you mean the 2004-5? In which case, it is not historical. Needs corrected throughout.
P4, line 5: Occupied rather than distributed?
P4, line 9: 2015-2016?
P4, line 11: Each year or just one year?
P4, line 13: What is the investigation route and investigation area? If you were to include statistical analysis, you could account for differences in site numbers in 2004/5 and 2015/16 which I am sure would show even more dramatic declines.
P4, line 17: Of birds, populations (not distributions).
P4, line 24: With a unique code is all you need to say.
P5, line 9: There is a real opportunity to show statistically significant differences between the two time periods and/or trends from 2004-2016 if the 2006-2012 data were included.
Table 1: What do the – mean? Not surveyed? There is considerable year to year fluctuation as shown particularly between 2015 and 2016. How do you account for this huge variation, presumably these birds are long-lived and the breeding success rate insufficient to account for such a huge increase? Or is this human error, in which case this is a serious concern.
Table 2: If the table headings were slightly more descriptive, there would be no need for the explanatory sentences thereafter.
P6, line 5 onwards: Some of this would be better presented as graphs.
Figure 3: Perhaps vary the colours as they are not discernible when printed in black and white.
P7: line 14: This is data from just four birds and should be summarised in fewer words.
P10: Line 11 onwards: This paragraph is repetition and not needed.
Discussion
P10, line 35: You need to be careful about inferring too much from the data of just four birds. I think further investigation is required to draw any conclusions about the results and usefulness of this technique for this particular species.
P11, line 1: What crises? I think we can assume breeding takes place in breeding areas.
P11, line 16: There are yet another two potential causes of decline but it is really not clear whether the causes are known. The discussion surrounding causes needs to be more concise, be clear about the evidence and consistent throughout the paper.
P11: line 35: Be more precise, what have the studies reported? Why is the data still limited? What is the relevance of GIS technology to do with submerged plants.
P11, line 50: Yes the abundance would drop if there was a rapid reduction in numbers! Reword or rephrase or simplify.
P11, line 52: Now ecological change. The authors need to be consistent in their terminology.
P12, line 3: What are the wintering risks?
P12, line 4: Is construction a threat? Why mention this?
P12, line 6: What do you mean by integrated assessments and valuations? There is a real opportunity here to recommend and direct the research towards understanding what is driving population changes/distributions but the text is not clear. E.g. line 10, yes perhaps more monitoring is needed but what is key is to include other factors such as submerged plant cover that would help to start to explain any changes.
P12, line 14: species’. Did it really provide more data than the observations?
P12, line 19: This paragraph should be shortened or removed. It does not add anything to the discussion.
P12, line 40: Broad statements that are not needed. This paragraph should be shortened simply to say that bird conservation should be as much, or more, about tackling species that are declining/endangered than inherently rare.
P13, line 1: Water birds?? Sentence does not make sense.
P13, line 5: Only four birds, keep results in perspective.
P13, line 16: Really not what future methods you are recommending/proposing. Perhaps a concluding paragraph in which you clearly and concisely describe these methods.
References
I don’t see the relevance of a reference on elephants for this paper.
Author Response
Thanks very much for the review! You are a patient, work carefully and responsibly person. You give me lots of useful suggestions. We thank you for your affirmation and support, and we can't finish our work without your help! It is very nice have such communication with you. We summarized major questions as below, thanks so much again!
Q1. There is no statistical analysis.
A: Thank you for your suggestion, we used Mann-Whitney U test to analysis the significance level of declines in Anhui and Poyang Lake between 10 years gap. The result is, is, count numbers showed a significant decline over the ten years in Anhui province as the p value lower than 0.05 (p=0.010). On the contrary, population declines in Poyang Lake were only marginally significant as the p value at 0.063 which lower than 0.1. For detailed interpret the population changes, we also provided mean counts and standard deviation of counted population numbers of A. cygnoides in the key sites. We put it in table 2 as part of our results. Please see Page 5, Line 24-37, Page 6, Line1-7 and Table 2.
Q2. I would recommend the authors simplify and streamline this paper focusing just on the population changes, backup by statistical analysis and concluding with a short discussion of possible causes and recommendations for further work.
A: Thank you for your suggestion, we have done the major revision by streamline on the population changes, we analyzed the significant level of declines by using statistic tools, and changed the structure of discussion and many descriptions in satellite tracking.
Q3. Abstract: Line 21: ‘Status of an ecosystem’. Is this the primary objective, what about conserving the bird itself?
A: Sorry for your confusion, we have totally changed the abstract by our newly revised paper. And we have removed this part in the abstract. Please see Page1, Line 16-31.
Q4. Abstract: Line 24: How does it contribute to environmental health?
A: Sorry for your confusion, we used to try to explain about build up a bio-indicator based monitoring system, which may contribute to maintain environmental health. But this topic is huge, thus we have totally changed the abstract by our newly revised paper. And we have removed this part in the abstract. Please see Page1, Line 16-31.
Q5. Abstract: Line 25: Clarifying the degree of ecosystem…… This sentence is confused.
A: Sorry for your confusion, we have totally changed the abstract by our newly revised paper. And we have removed this part in the abstract. Please see Page1, Line 16-31.
Q6. Abstract: Line 28: ‘…with satellite telemetry.’ To do what? For what purpose?
A: Sorry for your confusion, the satellite may be a robust tool as data mining, generate massive ecological data towards animal behavior. Those may contribute in abundance and distribution research. But we have totally changed the abstract by our newly revised paper. And we have removed this part in the abstract. Please see Page1, Line 16-31.
Q7. Abstract: Line 32: This is very speculative without data to back it up. It should be part of the discussion but not the abstract.
A: Thank you for your suggestion, we have removed it from the abstract. And we totally changed the abstract by our newly revised paper. Please see Page1, Line 16-31.
Q8. Abstract: Line 35: The aims of the study should not be at the end of the abstract, it should be the results/implications.
A: Thank you for your suggestion, we have totally rewritten our abstract based on our revised paper, please see Page1, Line 16-31.
Q9. Introduction: Line 4: This sentence does not make sense.
A: Thank you for your suggestion. We would like to say the waterbirds community are quick response to environmental changes due to their fast mobility, their distribution changes may reflex ecosystem degradation, thus they can be used as bioindicators in environmental monitoring system. We have rewritten the sentence and add more citations. Please see Page 2, Line 2-3.
Q10. Introduction: Line 5: The relevance of this statement is unclear.
A: Thank you for your suggestion. We add some explanations to show the relevance, for example, goose is large, abundant, and easy to identify in a long distance compare to various small ducks and rare species like cranes. Thus, the census data on goose are more feasible for environmental protectors. Please see Page 2, Line 4-7.
Q11. Introduction: Line 7: And conserving the birds…..?
A: Yes, there are many species conservation managers and species protectors. We have specialized the term conservation managers by species conservation managers. Please see Page 2, Line 6-7.
Q12. Introduction: Line 11: ‘basically distributed soley’ poor grammar.
A: Yes, we accept it, and we found this sentence is redundancy to the distribution introductions below, thus we have removed this sentence.
Q13. Introduction: Line 22: But you listed climate change in the abstract? Are the reasons really known or this all speculation? I think you need to be clearer and consistent.
A: Thank you for your suggestion, there are few studies relate waterbird distribution to hydraulic and temperature changes, but indeed, it is not clear about the influences of climate changes, thus we have changed it and also rewritten the abstract. Please see Page 2. Line 20-21.
Q14. Introduction: Line 27: What database?
A: Here we would like to introduce in some developed country, such as Canada, there are many websites created by NGOs, and the public can upload their observation data on species, such information may form a large database. We changed the word such a database to bird monitoring database. Please see Page 2 Line 26-33.
Q15. Introduction: Line 30: Is a culture of bird watching really something that is ‘developed’?
A: Sorry for your confusion. Here we addressed that culture of bird watching under development means people in China normally pay more attention on photo the bird than upload useful survey data, thus the contribution of citizen science towards species is under developing. Anyway, we have substituted this sentence. Please see Page 2 Line 26-33.
Q16. Introduction: Line 43: This paragraph appears to be suggesting that most bird data is collected by professionals in China and that the quality is dependent on how much they get paid. Whereas what I think you mean to say is that surveys are generally only carried out by professionals, of which there is a lack, and the time-consuming nature of them means new cost-effective methods are needed.
A: Yes, we would like to say currently the field survey needs put in a lot of money and labor. And the birds are highly mobile, thus quality of survey data is hard to maintain in high level. If the satellite tracking method could be proved to substitute part of the labor work, it would be more efficient and money-saving in the field survey. We rewrote some part of this paragraph for better understanding of readers. Please see Page 2, Line 34-38.
Q17. Methods: P3, line 14: ‘benefit A. cygnoides as a food source’.
A: Thank you for your suggestion. We have edited the sentence by your advice. Please see Page 3, Line 10.
Q18. Methods: P3, line 17: How have they degraded? Fewer species, invasive species, reduced abundance, biomass?
A: Thank you for your suggestion. The major reason is by the massive aquiculture and fish farming, we have included the reason to the original paragraph. Please see Page 3, Line 12-14.
Q19. Methods: P4, line 2: Of all sites? Where is the 2006-2012 data?
A: Thank you for your asking. We do have count data from 2006-2012. But the quality of count data from this time range is not good enough to present here due to deficiency in foundation, in some year we didn’t survey all the lakes in different provinces. That is why we only compared the data from 2004,2005,2015, and 2016. Those data comparatively higher in integrity. We changed the description for better understanding in the paragraph and add a table as appendix to show the complete count data from 2004-2016. Please see Page 3. Line 21-26.
Q20. Methods: P4, line 3: This sentence is not needed. And what historical data? I presume you mean the 2004-5? In which case, it is not historical. Needs corrected throughout.
A: Thank you for your suggestion, we have rewritten this part and add new appendix to show the complete survey data from 2004-2016. Please see Page 3. Line 21-26.
Q21. Methods: P4, line 5: Occupied rather than distributed?
A: Thank you for your asking, we rewritten this method part to shorten and make the description clearer. Please see Page 4, Line 1-3.
Q22. Methods: P4, line 9: 2015-2016?
A: Yes, in 2015 we surveyed 63 sub-lakes and in 2016 we surveyed 69 sub-lakes in Poyang lake. We rewritten this method part to shorten and make the description clearer. Please see Page 4, Line 6-8.
Q23. Methods: P4, line 11: Each year or just one year?
A: Thank you for your asking, the survey period is for each year, we do a pre-survey in the November, and the formal survey is from January to February. We rewritten this method part to shorten and make the description clearer. Please see Page 4, Line 3-4.
Q24. Methods: P4, line 13: What is the investigation route and investigation area?
A: Thank you for your asking and sorry for the confusion. Here the investigation route we mean the survey route from one observation site to another, and the investigation means Poyang lake. Anyway, we rewritten this method part to shorten and make the description clearer. Please see Page 4, Line 10-15.
Q25. Methods: P4, line 17: Of birds, populations (not distributions).
A: Thank you for your suggestion. We have improved it in the paper. Please see Page 4, Line 12-15.
Q26. Methods: P4, line 24: With a unique code is all you need to say.
A: Thank you for your suggestion and we accepted it. Normally each bird has two codes, one is on neck collar which is easy to be observe, and the other code is the code for each tracking logger. So, we eliminated the description about the neck collar and keep the code for tracking logger. Please see Page 4, Line 19-28.
Q27. Results: P5, line 9: There is a real opportunity to show statistically significant differences between the two time periods and/or trends from 2004-2016 if the 2006-2012 data were included.
A: Thank you for your suggestion, we use Mann-Whitney U test analyzed the difference of count numbers between the two time periods (2004 & 2005 and 2015 & 2016) in Anhui and Poyang Lake. Count number in every sub-lakes were tested. The result is, count numbers showed a significant decline over the ten years in Anhui province as the p value lower than 0.05 (p=0.010). On the contrary, population declines in Poyang Lake were only marginally significant as the p value at 0.063 which lower than 0.1. Please see Page 6, Line 3-7.
Q28. Results: Table 1: What do the – mean? There is considerable year to year fluctuation as shown particularly between 2015 and 2016.
A: Sorry for the confusion. The dash means data not available due to there is a deficiency of investigation in that province. We have change it to N.A.
The huge fluctuation between 2015 and 2016 due to many reasons, first of all, count data is not very stable because the bird is high in mobility, thus the same site in different time may cause huge fluctuation. Second, these two years surveyed by two different team, thus the human experiences on observation may be another reason. Anyway, this is the deficiency for counting data to present species population and distribution changes, and that is why we try to design another tracking method to increase the accuracy on data. Please see Page 5, Table 1.
Q29. Results: Table 2: If the table headings were slightly more descriptive, there would be no need for the explanatory sentences thereafter.
A: Thank you for your suggestion, we have changed the Table 2. Please see Page 6 Table 2.
Q30. Results: P6, line 5 onwards: Some of this would be better presented as graphs.
A: Thank you for your suggestion, but we think the first sentence here is more like a general introduction, thus we decide to delete it from the result part.
Q31. Results: Figure 3: Perhaps vary the colours as they are not discernible when printed in black and white.
A: Thank you for your suggestion and we accepted it, we have changed the color of the figure 3. Please see Page 6-7, Figure 3.
Q32. Results: P7: line 14: This is data from just four birds and should be summarised in fewer words.
A: Thank you for your suggestion, and we accept it. We have explained in few words about the four birds. Please see Page 7, Line 26-28.
Q33. Results:P10: Line 11 onwards: This paragraph is repetition and not needed.
A: Thank you for your suggestion and we accept it. We have removed it from the paper.
Q34. Discussion: P10, line 35: You need to be careful about inferring too much from the data of just four birds.
A: Thank you for your suggestion, and here we changed the description to a more general discussion. We discussed about the current importance of field survey and the possible application of satellite tracking in the future. Please see Page 10, Line4-12.
Q35. Discussion: P11, line 1: What crises? I think we can assume breeding takes place in breeding areas.
A: Sorry for the confusion. Here we tried to explain the sharp decline in numbers of Swan Goose population may inferring to decline of birth rate in breeding area and migration survival rate on migration way. But those are just inferring, thus, we decide remove these from discussion.
Q36. Discussion: P11, line 16: There are yet another two potential causes of decline but it is really not clear whether the causes are known. The discussion surrounding causes needs to be more concise, be clear about the evidence and consistent throughout the paper.
A: Thank you for your suggestion, we have improved our discussion, we better classified our topics in three parts including abundance change, distribution change, and satellite tracking. In this paragraph, we discussed more about distribution changes towards Figure 3. Please see Page 9, Line 1-42.
Q37. Discussion: P11: line 35: Be more precise, what have the studies reported? Why is the data still limited? What is the relevance of GIS technology to do with submerged plants.
A: Thank you for your suggestion, and we accept it. We would like to explain that, there are many studies carried on by different researchers, and they are normally stand on their own point of view, the studies on certain lakes also cannot cover the whole Poyang Lake, we have improved this discussion and integrated it to the upper paragraph. GIS technology has the potential to identify plant richness by the NDVI value, but it seems not work well for submerged plants, anyway, it is not highly relevant to this study and we have removed it.
Q38. Discussion: P11, line 50: Yes the abundance would drop if there was a rapid reduction in numbers! Reword or rephrase or simplify.
A: Thank you for your suggestion, and we accept it. We now simplified this sentence and integrated it to the first paragraph of discussion part. We believe this arrangement may show a better hierarchical structure of discussion as discuss about abundance changes, discuss about distribution changes, and discussion the application of satellite tracking. Please see Page 8-10.
Q39. Discussion: P11, line 52: Now ecological change. The authors need to be consistent in their terminology.
A: Thank you for your suggestion, and we accept it. We have change it to environmental changes, and integrated it to the first paragraph of discussion part. Please see Page 8, Line 11-26.
Q40. Discussion: P12, line 3: What are the wintering risks?
A: The wintering risks could be hunted in the Poyang Lake. We decide to remove this inferring from the paper due to lake of supportive evidences.
Q41. Discussion: P12, line 4: Is construction a threat? Why mention this?
A: Sorry for your confusion. At the beginning, we inferring the distribution changes caused by the dam construction in Poyang Lake, but we now remove it from discussion due to lack of supportive evidences.
Q42. Discussion: P12, line 6: What do you mean by integrated assessments and valuations?
A: Sorry for your confusion. At the beginning, we aim to explain about building up a bioindictor based ecosystem evaluation system, but the topic is too big and make the discussion tedious, thus we decide remove it to shorten the discussion
Q43. Discussion: P12, line 14: species’.did it really provide more data than the observations?
A: Thank you for your suggestion. Satellite tracking data can provide more activity data of specimen, but it is still hard to replace the observations currently. we now rewrite this part to discuss more on the potential of this technologies towards ours results. Please see Page 9-10, Line 43-52, 1-12.
Q44. Discussion: P12, line 19: This paragraph should be shortened or removed.
A: Thank you for your suggestion, and we accept it, we have removed it from the paper.
Q45. Discussion: P12, line 40: Broad statements that are not needed.
A: Thank you for your suggestion, and we accept it, we have removed it from the paper.
Q46. Discussion: P13, line 1: Water birds?? Sentence does not make sense.
A: Thank you for your suggestion,but we have to explain that waterbird is a term in ornithology, specific to birds rely on water, such as ducks, goose, and swan. There are many relevant papers used this term, for example: Cintra, 2019. Waterbird community composition in relation to lake physical traits and wetland limnological conditions in the Amazon basin. HYDROBIOLOGIA, 826(1):43-65. Anyway, we change the word water birds to waterbirds by eliminate the dash.
Q47. Discussion: P13, line 5: Only four birds, keep results in perspective.
A: Thank you for your suggestion, we have rewritten our conclusion to be more perspective. Please see Page 10, Line 14-30.
Q48. Discussion: P13, line 16: Really not what future methods you are recommending/proposing. Perhaps a concluding paragraph in which you clearly and concisely describe these methods.
A: Thank you for your suggestion, we have rewritten our conclusion to be more perspective. Please see Page 10, Line 14-30.
Q49. References: I don’t see the relevance of a reference on elephants for this paper.
A: The reference “de Boer et al, 2013” was to prove that “wild animal populations are sensitive to environmental change and habitat quality.” It is true and better that use some other citations related to waterbirds. We have removed the first sentence in the introduction part and rewrite it by cited other papers which study on the relationship between environmental changes and waterbirds. Please see Page 10, Line 40-45.

Round 2
Reviewer 1 Report
I found that the authors significantly improved the paper from the lat version, both in the text and tales and figures. The paper has been also improved by the introduction of statistical tests in the paper. As several parts of the paper have been rewritten I suggest the authors to re-read again the whole manuscript in order to (1) remove redundant information (persented for example both in the results and discussion) and (2) improve the clarity of the sentences by reducing their length and simplifying them. This will benefit the readers understanding and increase the interest of the audience for the paper. At this stage some parts of the paper are lengthy to read. Finally I ask to pay attention to the English form, by checking for verb tense consistency and word meaning. I recommend the acceptance of the paper after the last changes have been completed.

Author Response
Thanks very much for the review! We appreciate what you commented on our paper. You are a patient, work carefully and responsibly person. We thank you for your affirmation and help, and we can't finish our work without your help! It is very nice have such communication with you. We summarized major questions as below, thanks so much again!
Q1. remove redundant information (presented for example both in the results and discussion)
A: Thank you for your suggestion, we have removed all the redundant information through the article.
Q2. Improve the clarity of the sentences by reducing their length and simplifying them.
A: Thank you for your suggestion, we have carefully changed the long sentences by made them short and simple.

Reviewer 2 Report
I have unfortunately not seen a covering letter amongst the documents. That aside, I can see that the authors have made considerable changes to the paper based on the review comments. However, the changed text has poor quality English which makes the paper difficult to read and understand. I would recommend a revision being fully checked by a native English speaker before resubmission. I have tried to provide, I hope helpful, suggestions for text amendments below but in some cases cannot fully understand what is being said.
General comment
Confusing about the survey time periods. If I understand correctly the winter surveys ran from e.g. Dec 2015 to Feb 2016, and then Dec 2016 to Feb 2017. It would be better throughout therefore to refer to the survey periods as 2004/5 2005/6 and 2015/6 and 2016/7…..etc.
Abstract
Line 17 and Introduction line 2: ‘are quick to respond to’
Line 25 and Introduction line 22: ‘and monitoring procedures are both necessary’
Line 31: ‘as this may provide a substitute for traditional field surveys thus reducing the funds and labour costs required for winter field surveys’.
Introduction
Line 3: ‘Compared to’
Line 4: The birdwatchers are drawn to the birds not the other way around.
Line 6: ‘is relatively easy compared to other rare species and is crucial’ or ‘is easier than other rare species and is crucial’
Line 19: ‘A. cygnoides’
Line 24: ‘field surveys was the primary data’. I think most people would know that field surveys involve people and vehicles. I would remove from the next sentence.
Line 26: ‘In some developed countries (e.g. …..), bird monitoring databases are…..following well-organized, large-scale survey initiatives’.
Line 28: ‘In developing countries, such as China, the contribution of citizen science….’. I don’t know what is meant by specialization. Does it mean, people don’t have the skills, the interest, are not engaged in recording….etc, or is the problem with organisations leading initiatives and collated data? Or is it a combination of factors?
Line 32: Do you mean the length of time spent observing, or the seasonal timing of observations? It is unclear why site choice affects quality. Who is the long-term funding for – scientists, or organisations collating data?
Line 32: ‘and experience of observers’ or ‘and skill level of observers’.
Line 33: ‘statistical models’
Line 35: Unclear. If field protocols and statistics are being used to maximise the quality of the data then it is unclear why there is still an issue with accuracy and quality?
Line 35: ‘a modern technology-based survey method such’
Line 37: ‘in the field. This technique has already been widely used….’
Line 39: ‘from field surveys and compare….’ And line 41.
Line 43: ‘which may be applied to reduce the funding and, particularly, the labour costs required by winter field surveys.
Line 44: ‘may help further advance the conservation prospects of this species’.
Methods section 2.1
Line 15: Unclear what is meant by massive aquiculture? Does it cover a large area, large species, is it increasing, more intensive?
Line 16: ‘loss of some important…’
Methods section 2.2
Line 1: I presume you mean ‘due to the higher quality of the datasets’.
Line 3: This sentence is unclear.
Line 5: ‘The survey period was from December to mid-February the following year’. Do you mean former rather than formal surveys?
Line 7: At a similar time or date rather than period?
Line 8: ‘investigation site’.
Line 8: Why not just say ‘Sixty-three sub-lakes of Poyang lake were investigated in 2015 and 80 sub-lakes in 2016’.
Line 12: ‘into several discrete areas using clear geographic boundaries’.
Line 12: I presume you counted the number of birds rather than the number of populations?
Line 13: ‘and recorded using a monocular telescope. Count numbers and the geographic coordinates of observation sites were recorded’. The rest of the sentence is irrelevant.
Line 16: ‘To assess whether there were significant differences in bird numbers between the two time periods (2004/5 and 2015/16), two-year medians were analysed using a Mann-Whitney U-test’.
Methods section 2.3
Line 23: ‘Four birds arriving to winter in Poyang Lake were fitted with necklace collar type solar-powered GPS/GSM satellite transmitters from Ecotone Telemetry Company (ecotone-telemetry.com)’.
Line 25; ‘equipment was 44g, …..’
Line 2: Birds is a better word than specimens.
Line 6: ‘wintering site). We then created a satellite……’
Results section 3.1
Line 10: ‘The population size of A. cygnoides was estimated at approximately 61,000…’
Line 12: ‘winter field seasons, the population size of A. cygnoides was estimated at approximately 8,800 and 25,000 respectively. The vast majority of the population was concentrated….’
Line 15: What do you mean by average? Mean? You mentioned median before. This sentence needs reworded something along the lines ‘The mean population of the Yangtze River floodplain showed a 72% decline in the ten year period between 2004 and 2005, and 2015 and 2016’.
Line 16: Unclear. Is this 78% of the 72%?
Table 1: N.A. means Not Available reads as if there were no means available whereas I think you mean there were no data available. Not sure why Not Available are capitalised. In the table you have e.g. Average of 2015&2016 for Hubei but it is not an average of the years if there were no data available for 2015. It should be made clear in the text that if only one year of the two-year period was available you just used that value and not a mean.
Line 24: ‘for A. cygnoides…..’ Many readers may not understand what ‘Waterbirds Population Estimation 3’ means. Add a reference or explain at first mention.
Line 25: ‘where populations were estimated at 30,880 (in 2004) and 37,030 (in 2005).’
Line 25: ‘Nearly 98% of A. cygnoides….’
Line 26: Mean count number is a phrase that should be changed throughout, e.g. used mean bird numbers, or mean population estimates, or mean number of birds counted was…..etc.
Line 28: ‘were counted at those lakes with no sites identified as key sites of importance…..’
Line 30: ‘years, all lakes in Anhui Provence lost their designation ….’
Line 31: ‘for this species…’
Line 38: ‘a large proportion of the population of A. cygnoides was concentrated in 6….’
Line 1: I am not familiar with the term centralized rate. It is not necessary to have all the years in brackets.
Line 3: ‘(in 2004) to 3,528…..’
Line 4: This sentence is not needed, if you choose to keep it there are grammatical and spelling errors. You could simply add ‘as tested by a Mann-Whitney’ after Anhui province.
Line 6: The p value is shown so it unnecessary to also say it is lower than 0.05. Similar with next sentence.
Line 7: Arguably this is not significant. Why did you use the level of 0.1?
Table 2: Title needs to include the birds name otherwise this table is useless out of context. The legend of the table needs to be checked for English language, it currently does not make sense.
p.7, line 5/6: This sentence does not have a purpose. Suggest added ‘are shown in Figure 3’.
Line 7: Use supported rather than sustained.
Line 14: ‘showed a slight change’ or ‘showed slight changes’
Line 24: Is the word formidable necessary? I am missing the details now on when the birds were tagged.
p. 8, line 3: Sentence needs reworded. Possibly: ‘The arrival and departure dates in Poyang Lake was unique to each individual. The total wintering period of these four tracked A. cygnoides varied from 77 to 111 days during the 2015 winter and 105 to 146 days during the 2016 winter (Appendix 3).’
Line 8: ‘through the winters of 2015 and 2016…’
Line 15: I presume you mean ‘Apart from one missing specimen…’
Line 20: ‘the wintering period…’
Discussion
p. 9, line 7: ‘the primary research method for studying the abundance and distribution of A. cygnoides’.
Line 8: This sentence makes no sense, especially ‘digging out which behind those data’. Please check and reword.
Line 10: ‘As shown in Table 1,’
Line 11: ‘In agreement with other studies, Anhui and Jiangxi Provinces have become….’ Not sure what is meant by ‘shelter’, presumably you mean something like stronghold and not literally somewhere they shelter. Inconsistent throughout the paper whether province is capitalised or not.
Line 12: Changes to something like ‘More worrying is that lakes in Anhui province, which are former Ramsar wetlands, are losing their importance for wintering A. cygnoides.’.
Line 14: This sentence makes no sense.
Line 16: Change the words occasionality and randomicity, they make no sense in this context. It is really not clear what the authors are trying to say.
Line 17: ‘In addition,’. It is still unclear what the authors are trying to say here. Does the topography and vegetation, for example, make it difficult to see all the birds? What is about location that causes errors, what geomorphic observations are you making??
Line 20: I doubt it generated important sites, more probably it recognised important sites.
Line 22: Sentence does not make sense.
Line 27: Not common practice to refer to figures in the discussion. ‘We have shown that more A. cygnoides….’ but please check the English, this currently does not make sense.
Line 29: Sentence needs reworded, does not make sense.
Line 35: ‘Food availability may be the primary reason for distribution change for this tuber-feeding waterbird.’
p.10, line 1: ‘A case in 2015 and 2016’ does not make sense, please reword.
Line 2: Again, reference to figure. Should it not be mudflats?
Line 3: ‘attracted a large population’ or ‘attracted a large number’. Sentence starting ‘The Kangshan…’ does not make sense, please reword.
The remainder of the discussion needs to be checked for English. Unfortunately there are many places where I just do not understand your meaning. The same applies to the Conclusions.
Author Response
Thanks very much for the review! You are a patient, work carefully and responsibly person. You give me lots of useful suggestions. We thank you for your affirmation and support, and we can't finish our work without your help! It is very nice have such communication with you. We summarized major questions as below, thanks so much again!
Q1. Confusing about the survey time periods.
A: Sorry for your confusion. The survey time period is from Dec 2015 to Feb 2016, and then the next year survey start from Dec 2016 to Feb 2017. We will change it to 2004/5 and 2015/6 etc.
Q2. Abstract: Line 17 and Introduction line 2: ‘are quick to respond to’
A: Yes, we agree, and we have modified it. Please see Page 1-2, Line 17 and 2.
Q3. Abstract: Line 25 and Introduction line 22: ‘and monitoring procedures are both necessary’
A: Yes, we agree, and we have modified it. Please see Page 1-2, Line 26 and 23.
Q4. Abstract: Line 31: ‘as this may provide a substitute for traditional field surveys thus reducing the funds and labour costs required for winter field surveys’.
A: Yes, we agree, and we have modified it. Please see Page 1, Line 31-32.
Q5. Introduction: Line 3: ‘Compared to’
A: Yes, we agree, and we have modified it. Please see Page 2, Line 3.
Q6. Introduction: Line 4: The birdwatchers are drawn to the birds not the other way around.
A: Yes, we agree, and we have changed the order of sentence. Please see Page 2, Line 4-5.
Q7. Introduction: Line 6: ‘is relatively easy compared to other rare species and is crucial’ or ‘is easier than other rare species and is crucial’
A: Yes, we agree, and we have modified it. Please see Page 2, Line 6-7.
Q8. Introduction: Line 19: ‘A. cygnoides’
A: Yes, we agree, and we have modified it. Please see Page 2, Line 20.
Q9. Introduction: Line 24: ‘field surveys was the primary data’. I think most people would know that field surveys involve people and vehicles. I would remove from the next sentence.
A: Yes, we agree, and we have modified it. Please see Page 2, Line 25-27.
Q10. Introduction: Line 26: ‘In some developed countries (e.g. …..), bird monitoring databases are…..following well-organized, large-scale survey initiatives’.
A: Yes, we agree, and we have modified it. Please see Page 2, Line 27-30.
Q11. Introduction: Line 28: ‘In developing countries, such as China, the contribution of citizen science….’. I don’t know what is meant by specialization. Does it mean, people don’t have the skills, the interest, are not engaged in recording….etc, or is the problem with organisations leading initiatives and collated data? Or is it a combination of factors?
A: Sorry for your confusion. In China, amateur birdwatchers more like to take photos of birds instead of take a note of the watching place and location. Furthermore, the census data sharing between experts is also very low, and different research group may use different census form and give different notes. Thus, we mentioned specialization here means to develop a more powerful network for data sharing, and involved more amateur join in and provide more valuable data. We have modified the sentence, please see Page 2, Line 30.
Q12. Introduction: Line 32: Do you mean the length of time spent observing, or the seasonal timing of observations? It is unclear why site choice affects quality. Who is the long-term funding for – scientists, or organisations collating data?
A: Sorry for your confusion. Actually, the timing here means the occasion that the amount of birds were observed on the sites by observers. Because, the birds are highly mobility, it happened many times that when you get the observation site, those birds just leaving, and when you go to another site, those bird fly back. It may cause omit in census data. We have changed the investigation times to investigation timing to avoid omissions. Please see Page 2, Line 33-34.
Q13. Introduction: Line 32: ‘and experience of observers’ or ‘and skill level of observers’.
A: Yes, we agree, and we have modified it. Please see Page 2, Line 34.
Q14. Introduction: Line 33: ‘statistical models’
A: Yes, we agree, and we have modified it. Please see Page 2, Line 35.
Q15. Introduction: Line 35: Unclear. If field protocols and statistics are being used to maximise the quality of the data then it is unclear why there is still an issue with accuracy and quality?
A: Sorry for your confusion. Although there are some protocols and statistics are being used to maximize the quality of the data. But those protocols are normally developed from some single lakes. In Poyang Lake, there are many sub-lakes, and the geomorphology and climate conditions are different from lakes to lakes. Thus, even use the some protocol, the quality is still cannot guarantee in different situations. We added an explanation at the end as “due to different geomorphology and climate situations from lake to lake. Please see Page 2, Line 37-38.
Q16. Introdcution: Line 35: ‘a modern technology-based survey method such’
A: Yes, we agree, and we have modified it. Please see Page 2, Line 37.
Q17. Introduction: Line 37: ‘in the field. This technique has already been widely used….’
A: Yes, we agree, and we have modified it. Please see Page 2, Line 39.
Q18. Introduction: Line 39: ‘from field surveys and compare….’ And line 41.
A: Yes, we agree, and we have modified it. Please see Page 2, Line 41 and 43.
Q19. Introduction: Line 43: ‘which may be applied to reduce the funding and, particularly, the labour costs required by winter field surveys
A: Yes, we agree, and we have modified it. Please see Page 2, Line 45-46.
Q20. Introduction: Line 44: ‘may help further advance the conservation prospects of this species’.
A: Yes, we agree, and we have modified it. Please see Page 2, Line 46-47.
Q21. Methods 2.1: Line 15: Unclear what is meant by massive aquiculture? Does it cover a large area, large species, is it increasing, more intensive?
A: Sorry for your confusion. The massive aquiculture here means the people there usually use a large area of lake for pearl culture, and during autumn, they harvest fishes intensively.
Q22. Methods 2.1: Line 16: ‘loss of some important…’
A: Yes, we agree, and we have modified it. Please see Page 3, Line 17.
Q23. Metnods 2.2: Line 1: I presume you mean ‘due to the higher quality of the datasets’.
A: Yes, we agree, and we have modified it. Please see Page 4, Line 3.
Q24. Methods 2.2: Line 3: This sentence is unclear.
A: Sorry for your confusion. We re-designed the observation sites in the newly investigation surveys in 2015 and 2016. More observation sites were included based on the former investigation in 2004 and 2005. Some important ramsar sites were also included in the newly surveys.
Q25. Methods 2.2: Line 5: ‘The survey period was from December to mid-February the following year’. Do you mean former rather than formal surveys?
A: Yes, we agree, and we have modified it. Please see Page 4, Line 7-8. Sorry for your confusion, normally there will be a pre-investigation in December, and the survey in January and February were formal. We decide to remove this sentence.
Q26. Methods 2.2: Line 7: At a similar time or date rather than period?
A: Sorry for your confusion. Date it was, we have change the world in the paper, please see Page 4, Line 9.
Q27. Methods 2.2: Line 8: ‘investigation site’.
A: Yes, we agree, and we have modified it. Please see Page 4, Line 10.
Q28. Methods 2.2: Line 8: Why not just say ‘Sixty-three sub-lakes of Poyang lake were investigated in 2015 and 80 sub-lakes in 2016’.
A: Yes, we agree, and we have modified it. Please see Page 4, Line 11-12.
Q29. Methods 2.2: Line 12: ‘into several discrete areas using clear geographic boundaries’.
A: Yes, we agree, and we have modified it. Please see Page 4, Line 14.
Q30. Methods 2.2: Line 12: I presume you counted the number of birds rather than the number of populations?
A: Yes, we counted the individual numbers in a population, we have changed it to avoid misunderstanding. Please see Page 4, Line 15.
Q31. Methods 2.2: Line 13: ‘and recorded using a monocular telescope. Count numbers and the geographic coordinates of observation sites were recorded’. The rest of the sentence is irrelevant.
A: Yes, we agree, and we have modified it. Please see Page 4, Line 16-17.
Q32. Methods 2.2: Line 16: ‘To assess whether there were significant differences in bird numbers between the two time periods (2004/5 and 2015/16), two-year medians were analysed using a Mann-Whitney U-test’.
A: Yes, we agree, and we have modified it. Please see Page 4, Line 19-22.
Q33. Methods 2.3: Line 23: ‘Four birds arriving to winter in Poyang Lake were fitted with necklace collar type solar-powered GPS/GSM satellite transmitters from Ecotone Telemetry Company (ecotone-telemetry.com)’.
A: Yes, we agree, and we have modified it. Please see Page 4-5, Line 27 and 1.
Q34. Methods 2.3: Line 25; ‘equipment was 44g, …..’
A: Yes, we agree, and we have modified it. Please see Page 5, Line 2.
Q35. Methods 2.3: Line 2: Birds is a better word than specimens.
A: Thank you for your advice, and we accept it. Please see Page 5, Line 7.
Q36. Methods 2.3: Line 6: ‘wintering site). We then created a satellite……’
A: Yes, we agree, and we have modified it. Please see Page 5, Line 10.
Q37. Results 3.1: Line 10: ‘The population size of A. cygnoides was estimated at approximately 61,000…’
A: Yes, we agree, and we have modified it. Please see Page 5, Line 14.
Q38. Results 3.1: Line 12: ‘winter field seasons, the population size of A. cygnoides was estimated at approximately 8,800 and 25,000 respectively. The vast majority of the population was concentrated….’
A: Yes, we agree, and we have modified it. Please see Page 5, Line 17-18.
Q39. Results 3.1: Line 15: What do you mean by average? Mean? You mentioned median before. This sentence needs reworded something along the lines ‘The mean population of the Yangtze River floodplain showed a 72% decline in the ten year period between 2004 and 2005, and 2015 and 2016’.
A: Sorry for your confusion. We only compared the data from two years, thus the medians and mean value are the same, and we have changed the medians to mean value in the method parts to avoid misunderstandings. Please see Page 4, Line 21.
Q40. Results 3.1: Line 16: Unclear. Is this 78% of the 72%? .
A: Yes, we mean what as you say. The calculation is (34,358-129)/(61,032-17,067)*100%
Q41. Results 3.1: Table 1: N.A. means Not Available reads as if there were no means available whereas I think you mean there were no data available. Not sure why Not Available are capitalised. In the table you have e.g. Average of 2015&2016 for Hubei but it is not an average of the years if there were no data available for 2015. It should be made clear in the text that if only one year of the two-year period was available you just used that value and not a mean.
A: Yes, N.A. means non data available, we directly used the value if there was 1 year data missing during average calculation. We have explained it in the text. Please see Page 5, Table 1.
Q42. Results 3.1: Line 24: ‘for A. cygnoides…..’ Many readers may not understand what ‘Waterbirds Population Estimation 3’ means. Add a reference or explain at first mention.
A: Yes, we agree, and we have modified it. Please see Page 5, Line 32,33 and 38.
Q43. Results 3.1: Line 25: ‘where populations were estimated at 30,880 (in 2004) and 37,030 (in 2005).’
A: Yes, we agree, and we have modified it. Please see Page 5, Line 33-34.
Q44. Results 3.1: Line 25: ‘Nearly 98% of A. cygnoides….’
A: Yes, we agree, and we have modified it. Please see Page 5, Line 34.
Q45. Results 3.1: Line 26: Mean count number is a phrase that should be changed throughout, e.g. used mean bird numbers, or mean population estimates, or mean number of birds counted was…..etc.
A: Yes, we agree, and we have modified it. Please see Page 5, Line 35.
Q46. Results 3.1: Line 28: ‘were counted at those lakes with no sites identified as key sites of importance…..’
A: Yes, we agree, and we have modified it. Please see Page 5, Line 37-38.
Q47. Results 3.1: Line 30: ‘years, all lakes in Anhui Provence lost their designation ….’
A: Yes, we agree, and we have modified it. Please see Page 5, Line 39.
Q48. Results 3.1: Line 31: ‘for this species…’
A: Yes, we agree, and we have modified it. Please see Page 6, Line 1.
Q49. Results 3.1: Line 38: ‘a large proportion of the population of A. cygnoides was concentrated in 6….’
A: Yes, we agree, and we have modified it. Please see Page 6, Line 8-9.
Q50. Results 3.1: Line 1: I am not familiar with the term centralized rate. It is not necessary to have all the years in brackets.
A: Sorry for your confusion. The centralization rate represent the amount of birds centralized in small place. And we will remove the years from the brackets. Please see Page 6, Line 8-10.
Q51. Results 3.1: Line 3: ‘(in 2004) to 3,528…..’
A: Thank you for your suggestion, we have changed it.
Q52. Results 3.1: Line 4: This sentence is not needed, if you choose to keep it there are grammatical and spelling errors. You could simply add ‘as tested by a Mann-Whitney’ after Anhui province.
A: Thank you for your suggestion, we have changed it. Please see Page 6, Line 13-14.
Q53. Results 3.1: Line 6: The p value is shown so it unnecessary to also say it is lower than 0.05. Similar with next sentence.
A: Thank you for your suggestion. But we think it is better to show the significant standard (0.05) here.
Q54. Results 3.1: Line 7: Arguably this is not significant. Why did you use the level of 0.1?
A: Yes, it is not significant, it just marginally significant. Thank you for your suggestion, we have removed the level of 0.1. Please see Page 6, Line 17.
Q55: Results 3.1: Table 2: Title needs to include the birds name otherwise this table is useless out of context. The legend of the table needs to be checked for English language, it currently does not make sense.
A: Thank you for your suggestion. We have changed the title of Table 2. Please see Page 6, Line 19-21.
Q56. Results 3.1: p.7, line 5/6: This sentence does not have a purpose. Suggest added ‘are shown in Figure 3’.
A: Thank you for your suggestion. We have changed it, please see Page 7, Line 5-6.
Q57. Results 3.1: Line 7: Use supported rather than sustained.
A: Thank you for your suggestion. We have changed it, please see Page 7, Line 7.
Q58. Results 3.1: Line 14: ‘showed a slight change’ or ‘showed slight changes’
A: Thank you for your suggestion. We have changed it, please see Page 7, Line 14-15.
Q59. Results 3.1: Line 24: Is the word formidable necessary? I am missing the details now on when the birds were tagged.
A: Thank you for your suggestion, now we changed the sentence and added the tag details. Please see Page 7, Line 24-25.
Q60. Results 3.2: p. 8, line 3: Sentence needs reworded. Possibly: ‘The arrival and departure dates in Poyang Lake was unique to each individual. The total wintering period of these four tracked A. cygnoides varied from 77 to 111 days during the 2015 winter and 105 to 146 days during the 2016 winter (Appendix 3).’
A: Thank you for your suggestion. We have changed it, please see Page 8, Line 4-8.
Q61. Results 3.2: Line 8: ‘through the winters of 2015 and 2016…’
A: Yes, we agree, and we have modified it. Please see Page 8, Line 8-9.
Q62. Results 3.2: Line 15: I presume you mean ‘Apart from one missing specimen…’
A: Yes, we agree, and we have modified it. Please see Page 8, Line 15.
Q63. Results 3.2: Line 20: ‘the wintering period…’
A: Yes, we agree, and we have modified it. Please see Page 8, Line 20.
Q64. Discussion: p. 9, line 7: ‘the primary research method for studying the abundance and distribution of A. cygnoides’.
A: Yes, we agree, and we have modified it. Please see Page 9, Line 6-7.
Q65. Discussion: Line 8: This sentence makes no sense, especially ‘digging out which behind those data’. Please check and reword.
A: Thank you for your suggestion, we have changed the sentence to “ species population and distribution changes may reveal deeper consequences related to environmental changes.” Please see Page 9, Line 8-10.
Q66. Discussion: Line 10: ‘As shown in Table 1,’
A: Yes, we agree, and we have modified it. Please see Page 9, Line 9.
Q67. Discussion: Line 11: ‘In agreement with other studies, Anhui and Jiangxi Provinces have become….’ Not sure what is meant by ‘shelter’, presumably you mean something like stronghold and not literally somewhere they shelter. Inconsistent throughout the paper whether province is capitalised or not.
A: Yes, we agree, and we have modified the word “shelter” by habitat areas. And we check the whole paper, and we uniformed the word as “province”. Please see Page 9, Line 11-13.
Q68. Discussion: Line 12: Changes to something like ‘More worrying is that lakes in Anhui province, which are former Ramsar wetlands, are losing their importance for wintering A. cygnoides.’.
A: Yes, we agree, and we have modified it. Please see Page 9, Line 13-14.
Q69. Discussion: Line 14: This sentence makes no sense.
A: Yes, we agree, and we have removed it from the paper. And changed the sentence to “Census-based survey data may provide quantity of data, but improve the quality is still needed.” Please see Page 9, Line 15-17.
Q70. Discussion: Line 16: Change the words occasionality and randomicity, they make no sense in this context. It is really not clear what the authors are trying to say.
A: Yes, we agree, and we have changed it. Please see Page 9, Line 18-19.
Q71. Discussion: Line 17: ‘In addition’. It is still unclear what the authors are trying to say here. Does the topography and vegetation, for example, make it difficult to see all the birds? What is about location that causes errors, what geomorphic observations are you making??
A: Sorry for your confusion. Sometimes, the observation sites were at low lands, thus the geese behind the high grass is very hard to discover. We have deleted this sentence to eliminate misunderstanding.
Q72. Discussion: Line 20: I doubt it generated important sites, more probably it recognised important sites.
A: Yes, we agree, and we have changed it. Please see Page 9, Line 23.
Q73. Discussion: Line 22: Sentence does not make sense.
A: Sorry for your confusion. This sentence do not very represented the meaning, we want to explain, the northeast used to be an important wintering place for swan geese before 2012. We have deleted this sentence.
Q74. Discussion: Line 27: Not common practice to refer to figures in the discussion. ‘We have shown that more A. cygnoides….’ but please check the English, this currently does not make sense.
A: Yes, we agree, and we have changed it. Please see Page 9, Line 30-32.
Q75. Discussion: Line 29: Sentence needs reworded, does not make sense.
A: Sorry for your confusion, we have changed the sentence. Please see Page 9, Line 32-33.
Q76. Discussion: Line 35: ‘Food availability may be the primary reason for distribution change for this tuber-feeding waterbird.’
A: Yes, we agree, and we have changed it. Please see Page 9, Line 39-40.
Q77. Discussion: p.10, line 1: ‘A case in 2015 and 2016’ does not make sense, please reword.
A: Sorry for your confusion, we have changed the sentence to “High water level may also influence on food availability for A. cygnoides.” Please see Page 10, Line 5-6.
Q78. Discussion: Line 2: Again, reference to figure. Should it not be mudflats?
A: We have changed those sentence. Please see Page 10, Line 6-11.
Q79. Discussion: Line 3: ‘attracted a large population’ or ‘attracted a large number’. Sentence starting ‘The Kangshan…’ does not make sense, please reword.
A: We have rewrote this part. Please see Page 10, Line 6-12.

Round 3
Reviewer 2 Report
The authors changes to this paper have improved it significantly. There are some minor English grammar/spelling issues which could now be handled by the editor.
Author Response
Thanks very much for the review! We appreciate what you commented on our paper. You are a patient, work carefully and responsibly person. We thank you for your affirmation and help, and we can't finish our work without your help! It is very nice have such communication with you. We summarized major questions as below, thanks so much again!
Q1. The authors changes to this paper have improved it significantly.
A: We appreciate your acceptation, we will keep trying and press on.
Q2. There are some minor English grammar/spelling issues which could now be handled by the editor.
A: Thank you for your suggestion, as you know our native language is Chinese, it is quite different language system to English, we will keep practice on English writing skills, but this time we will consider to ask a native English speaker to improve our paper.
